# PAF1c links S-phase progression to immune evasion and MYC function in pancreatic carcinoma

Abdallah Gaballa [1,8], Anneli Gebhardt-Wolf[1,8], Bastian Krenz [1,2], Greta Mattavelli[2], Mara John[2], Giacomo Cossa[1], Silvia Andreani[1], Christina Schülein-Völk[3], Francisco Montesinos[1], Raphael Vidal[1,4], Carolin Kastner[2], Carsten P. Ade [1], Burkhard Kneitz[5], Georg Gasteiger [6], Peter Gallant[1], Mathias Rosenfeldt [7], Angela Riedel[2] & Martin Eilers [1,4] ✉

In pancreatic ductal adenocarcinoma (PDAC), endogenous MYC is required for S-phase progression and escape from immune surveillance. Here we show that MYC in PDAC cells is needed for the recruitment of the PAF1c transcription elongation complex to RNA polymerase and that depletion of CTR9, a PAF1c subunit, enables long-term survival of PDAC-bearing mice. PAF1c is largely dispensable for normal proliferation and regulation of MYC target genes. Instead, PAF1c limits DNA damage associated with S-phase progression by being essential for the expression of long genes involved in replication and DNA repair. Surprisingly, the survival benefit conferred by CTR9 depletion is not due to DNA damage, but to T-cell activation and restoration of immune surveillance. This is because CTR9 depletion releases RNA polymerase and elongation factors from the body of long genes and promotes the transcription of short genes, including MHC class I genes. The data argue that functionally distinct gene sets compete for elongation factors and directly link MYC-driven S-phase progression to tumor immune evasion.

Expression of the MYC protooncogene is deregulated and elevated in a large proportion of pancreatic ductal adenocarcinoma (PDAC) tumors[1,2]. In approximately 10% of PDAC tumors, this is due to amplification of MYC; in the remaining cases, MYC expression is elevated because it is induced downstream of activating RAS mutations and because loss-of-function mutations in the TP53 and SMAD4 tumor suppressor genes abolish the suppression of MYC expression by the respective wild-type proteins[3,4]. Transgenic PDAC models driven by mutant KRAS and mutant p53 ("KPC") consequently express high levels of MYC and endogenous MYC is required for both tumor development and maintenance in these models[5–8]. Depletion of MYC in KPC models results in two major phenotypes: first, it causes a delay in cell cycle progression, which is most pronounced during S-phase[7]. Second, MYC is critical for preventing tumor eradication by the host immune system and the tumor regression observed in PDAC models upon MYC inactivation depends on an intact immune system[7,9,10].

[1]Department of Biochemistry and Molecular Biologyy, Theodor Boveri Institute, Biocenter, Julius Maximilian University Würzburg, Am Hubland, 97074 Würzburg, Germany. [2]Mildred Scheel Early Career Center, University Hospital Würzburg, Josef-Schneider-Str. 2, 97080 Würzburg, Germany. [3]Core Unit High-Content Microscopy, Theodor Boveri Institute, Biocenter, Julius Maximilian University Würzburg, Am Hubland, 97074 Würzburg, Germany. [4]Comprehensive Cancer Center Mainfranken, University Hospital Würzburg, Josef-Schneider-Str. 2, 97080 Würzburg, Germany. [5]Department of Urology and Pediatric Urology, University Hospital Würzburg, Josef-Schneider-Str. 2, 97080 Würzburg, Germany. [6]Würzburg Institute of Systems Immunology, Max Planck Research Group, Julius Maximilian University Würzburg, Versbacher Str. 9, 97078 Würzburg, Germany. [7]Institute of Pathology, Julius Maximilian University Würzburg, Josef-Schneider-Str. 2, 97080 Würzburg, Germany. [8]These authors contributed equally: Abdallah Gaballa, Anneli Gebhardt-Wolf. ✉e-mail: martin.eilers@biozentrum.uni-wuerzburg.de

MYC is a nuclear protein that exerts both global and gene-specific effects on transcription by all three RNA polymerases[11–13]. To exert these effects, MYC participates in several protein/protein complexes: the best characterized is a complex with MAX, which enables MYC to bind specific DNA sequences, known as E-boxes, and regulate the transcription of a wide range of target genes, most of which encode proteins involved in translation, metabolism and cell cycle progression[13]. Many current attempts to target MYC for therapy are therefore based on the hypothesis that the oncogenic function of MYC is a consequence of deregulated expression of its target genes and that the ability to regulate transcription from E-boxes is the crucial oncogenic function of MYC that needs to be inhibited[14,15]. However, recent proteomic analyses have identified a surprisingly complex interactome of MYC proteins[16–20] and have uncovered unanticipated functions of specific protein complexes in maintaining the genomic stability of tumor cells[12,20–22], raising the need to identify MYC´s oncogenic functions in an unbiased manner.

Both MYC and its neuronally-expressed paralogue, MYCN, can prevent transcription-replication conflicts (TRCs) and limit replication-associated DNA damage[20,21,23,24], suggesting that TRCs or DNA damage arising during S-phase cause the slow progression of MYC-depleted PDAC cells through S-phase and that factors that prevent TRCs or limit the DNA damage during S-phase may be critical for tumor growth. To test this hypothesis, we performed a focused siRNA screen searching for MYC cofactors that limit DNA damage during the S-phase of PDAC cells. Subsequent analyses led to the surprising conclusion that DNA replication and repair genes appear to compete for transcription elongation factors with genes encoding MHC class I proteins, which control the recognition of tumor cells by the host immune system, and that a transcription elongation complex termed PAF1c (RNA Polymerase associated complex)[25,26] controls the distribution of RNA Polymerase and elongation factors between both groups of genes. Consequently, PAF1c is required for both the prevention of DNA damage during S-phase and for MYC-dependent immune evasion. Since PAF1c is dispensable for the activation of canonical MYC target genes, our findings argue that targeting specific complexes and interaction partners can block MYC's oncogenic functions but spare its physiological functions in normal cell growth and proliferation.

## Results

### Depletion of MYC causes replication-transcription conflicts in PDAC cells

To study the function of endogenous MYC in PDAC cells, we expressed shRNAs targeting *MYC* in cells that have been derived from a tumor arising in the KPC mouse model driven by KRAS and p53 mutations (Fig. 1a)[5,7]. Depletion of MYC severely retards the proliferation of these cells. While progression through all phases of the cell cycle is delayed, the effects are most pronounced in S-phase, which extends from an average length of 4 to 12 h[7]. We performed several experiments to better characterize this phenotype: first, we showed that EdU incorporation is reduced upon MYC depletion, demonstrating that overall DNA replication is reduced (Fig. 1b; Supplementary Fig. 1a). Our parallel work had shown that depletion of MYC has no effect on the speed of unperturbed replication forks, but inhibits the recovery of replication of stalled forks, for example due to adjacent double-strand breaks[20]. Consistently, depletion of MYC led to a strong increase in the phosphorylation of gamma-H2AX (γ-H2AX), a marker of double-strand break formation, which occurred predominantly in the S-phase of the cell cycle (Fig. 1b–d; Supplementary Fig. 1a). Both MYC and its neuronal paralog, MYCN, have been implicated in the resolution of TRCs, suggesting that double-strand breaks upon MYC depletion may result from these conflicts[20,21,23]. To test whether TRCs occur in PDAC cells, we used proximity-ligation assays to measure the proximity of RNA polymerase II (RNAPII) with the proliferating cell antigen (PCNA), which forms the sliding clamp of active replication forks, and with

RAD9, which replaces PCNA at stalled replication forks as part of the 9-1-1 complex[27], using appropriate single-antibody controls. Both assays showed significant increases in the number of TRCs upon MYC depletion, providing evidence that endogenous MYC prevents TRCs (Fig. 1e). The stability of stalling replication forks depends on the ATR kinase, and ATR inhibition causes replication fork collapse and subsequent double-strand breaks[28]. Consistently, incubation of PDAC cells with AZD6738, a specific ATR inhibitor[29], caused a moderate increase in phosphorylation of KAP1 at serine 824, a substrate of the ATM kinase that is activated by double-strand breaks, in control cells, but caused a much stronger increase in MYC-depleted cells (Fig. 1f). BLISS sequencing of double-strand breaks provided direct evidence for a significant increase in the number of double-strand breaks in MYC-depleted cells treated with AZD6738 (Fig. 1g). Consistent with previous data, a large fraction of double-strand breaks occurred in transcribed regions close to promoters and depletion of MYC, either alone or in conjunction with ATR inhibition, enhanced the frequency of breaks both in transcribed and in intergenic regions (Supplementary Fig. 1b)[20,22]. Taken together, these data show that endogenous MYC reduces the frequency of TRCs in PDAC cells and limits the accumulation of double-strand breaks during S-phase progression.

### The PAF1 complex is critical for limiting DNA damage during S-phase

To identify critical co-factors that enable MYC to carry out these processes, we performed a focused siRNA screen of 86 factors that are either physically associated with MYC or MYCN or that are involved in processes that prevent DNA damage during S-phase such as R-loop resolution, polyadenylation, and promoter-proximal transcription termination (Supplementary Fig. 2a). For this screen, we used high-content automated microscopy analyzing three different parameters (Fig. 2a, b, Supplementary Fig. 2b and Supplementary Data 1). First, we looked for siRNAs that reduced the intensity of EdU incorporation during S-phase, indicating that they caused a delay in S-phase progression. Second, we monitored for an increase in KAP1 phosphorylated at serine 824, indicative of double-strand breaks[30], and third we determined whether low concentrations of AZD6738 increase KAP1(S824) phosphorylation upon factor depletion in a super-additive or synergistic manner. Each read-out identified between 8 and 56 significant hits (Fig. 2b, Supplementary Fig. 2c). Of these, four hits were positive in all three assays and a further eight hits were positive in two out of three assays (Fig. 2b). We noted that three subunits of the PAF1 complex (PAF1c), CTR9, CDC73 and RTF1, scored as hits and therefore decided to focus on the analysis of this complex in subsequent experiments. We validated the results of the screen using doxycycline-inducible shRNAs targeting each subunit and found that depletion of each of them, as well as depletion of MYC, strongly increased double-strand break formation in response to incubation with low doses of the ATR inhibitor AZD6738 (Fig. 2c). Virtually identical results were obtained upon depletion of CTR9, a central scaffold of PAF1c that is required for its assembly[31], in an additional p53-mutant and in a p16[ink4a](Cdkn2a)-deficient murine PDAC cell line, as well as in human (PA-TU-8988T) PDAC cells, arguing that the role of PAF1c is conserved (Supplementary Fig. 2d). Consistently, depletion of CTR9 had only a moderate effect on cell proliferation of PDAC cells, but completely suppressed proliferation of PDAC cells in the presence of low dose AZD6738 (Fig. 2d). Profiling a number of compounds that induce DNA damage or interfere with checkpoint responses showed that depletion of CTR9 sensitized KPC cells to compounds that interfere with the synthesis of thymidine and deoxynucleotides (5-FU, hydroxyurea), cause replication stress (cisplatin) and inhibit the CHK1 kinase that senses replication stress (CHK1: LY-2603618; CHK1 + 2: AZD-7762), but not to, for example, topoisomerase II inhibitors (doxorubicin,etoposide) or transcription-associated cyclin-dependent kinase inhibitors (CDK9: NVP2, CDK12: SR4835) (Fig. 2e and Supplementary Data 2).

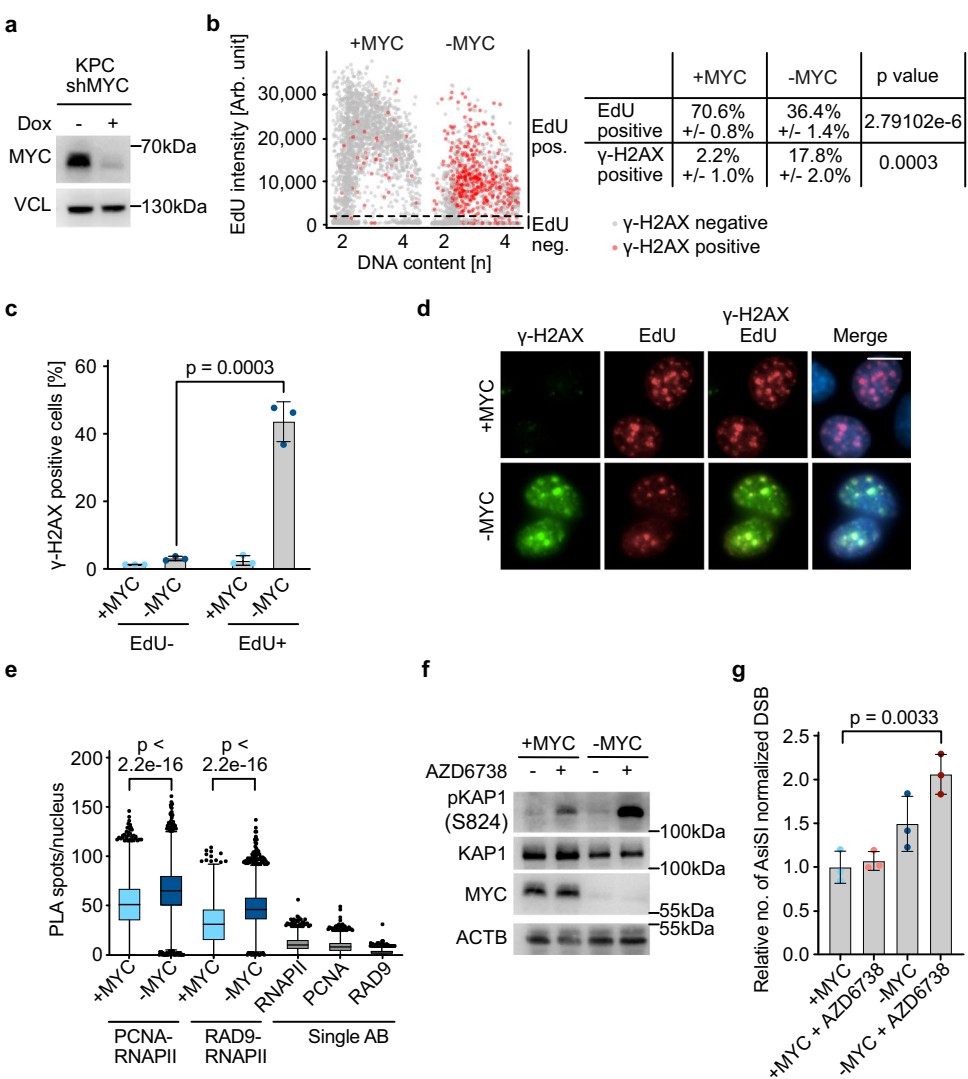

**Fig. 1 | Endogenous MYC prevents transcription-replication conflicts in PDAC cells. a** Immunoblot showing MYC expression in KPC cells. Where indicated, doxycycline (1 μg/ml) was added for 48 h. Vinculin was loading control (*n* = 3; *n* is number of independent experiments unless otherwise stated). **b** Quantitative image-based cytometry showing DNA content (Hoechst) on the *x*-axis and EdU incorporation on the *y*-axis of cells carrying doxycycline-inducible shRNA targeting *MYC*. "-MYC" indicates samples in which MYC depletion was induced by doxycycline addition (1 μg/ml) for 48 h and "+MYC" ethanol-treated control samples. EdU was added 30 min before fixation. For each independent replicate, 1000 cells were randomly selected and shown in this plot (*n* = 3). The table on the right shows mean values, standard deviations, and *P* values (unpaired two-sided *t*-test). **c** Bar plot showing the percentage of γ-H2AX positive cells in EdU⁻ or EdU⁺ cells, derived from (**b**). Data are presented as mean ± s.d. (*n* = 3; unpaired two-sided *t*-test). **d** Representative immunofluorescence images used for the analysis shown in (**b**). The merged image includes all stains including Hoechst to mark nuclei. Scale bar: 10 μm. **e** Single-cell quantification of nuclear PLA foci between either PCNA or RAD9

and total RNAPII. The following total number of cells was examined over three independent experiments: +MYC: PCNA-RNAPII *n* = 7715; −MYC: PCNA-RNAPII *n* = 2794; +MYC: RAD9-RNAPII *n* = 4934; −MYC: RAD9-RNAPII *n* = 3768; two-tailed Mann−Whitney *U*-test). Single antibody controls are shown for the +MYC condition (number of cells analyzed in one experiment: RNAPII: *n* = 2451; PCNA: *n* = 3232; RAD9: *n* = 2767). In the box plot, the central line shows the median and the borders of the boxes extend from the 25th to the 75th percentile, and the whiskers were plotted using the Tukey method and outliers are shown as black dots. **f** Immunoblot of cells harboring doxycycline-inducible shRNA targeting *MYC*. MYC depletion was induced as in (**a**) and AZD6738 (0.3 μM) was added for 48 h. Beta-actin was loading control (*n* = 3). **g** Bar plot showing the relative number of AsiSI-normalized reads, quantifying double-strand breaks as determined by BLISS sequencing. Doxycycline was added for 48 h to deplete MYC and AZD6738 (2 μM) for 2 h was added where indicated. Data are presented as mean ± s.d. (*n* = 3; unpaired two-sided *t*-test). Source data are provided as a Source Data file.

Finally, PLA assays showed that depletion of CTR9 or CDC73 had no significant effect on the proximity of RAD9 with RNAPII, arguing that it has no direct role in resolving TRCs (Supplementary Fig. 2e). Collectively, the data argue that PAF1c limits DNA damage predominantly during S-phase.

To test whether this protective function of PAF1C is related to MYC levels, we generated MYC-ER cells expressing inducible shCTR9. This showed that, as previously observed in U20S cells, MYC activation increased the percentage of EdU incorporating cells and depletion of CTR9 strongly decreased this percentage if MYC-ER had been

activated by 4-OHT but did not affect the overall percentage of EdU incorporating cells when MYC-ER had not been activated (Supplementary Fig. 3a). Furthermore, MYC-ER activation exacerbated DNA damage responses to depletion of CTR9: this was evidenced by an increase in CHK1 phosphorylation, a downstream target of ATR, as well as by increases in KAP1 and γ-H2AX phosphorylation, downstream targets of the ATM kinase (Supplementary Fig. 3b, c). Since ATM is activated by double-strand breaks, this indicated that CTR9 is required both for preventing MYC-enhanced replication stress and double-strand break formation. Furthermore, CTR9 depletion moderately

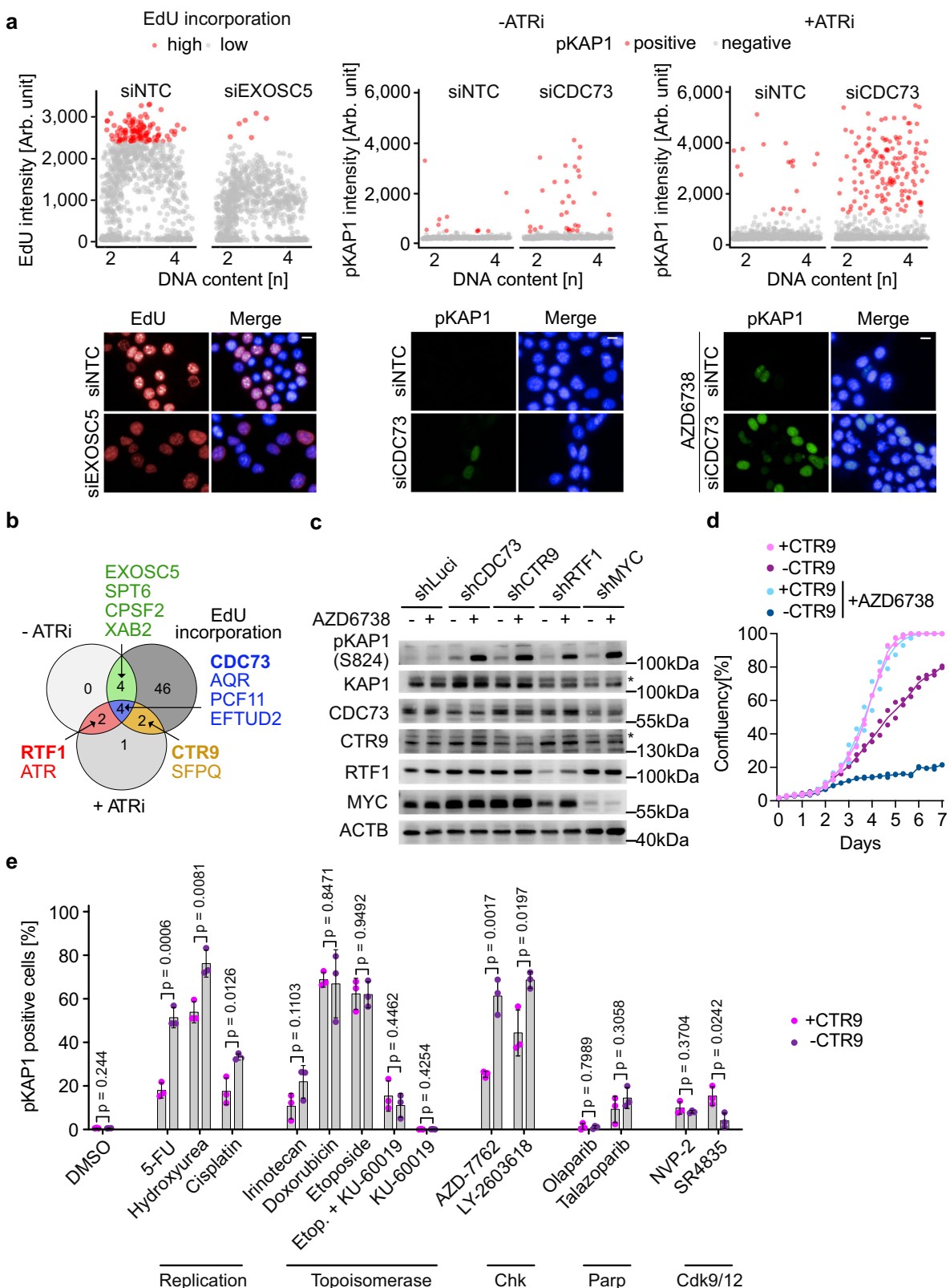

enhanced MYC-dependent apoptosis (Supplementary Fig. 3d). To alleviate replication stress, MYC can form multimeric, sphere-like structures and several PAF1C subunits are required for MYC multimerization in U2OS cells[20]. Consistent with these observations, depletion of CTR9 reduced sphere formation of MYC in KPC cells (Supplementary Fig. 3e). Finally, we used publicly available DEPMAP data[32] to correlate the dependence of multiple human pancreatic tumor cells on each of the genes identified as hits in the screen with MYC expression levels and found a strong correlation between higher level of MYC expression and high level of MYC target gene expression and the dependence on CTR9 and other subunits of the PAF1 complex, while the dependence of human PDAC cells on other genes was not (or anti-) correlated with MYC levels (Supplementary Fig. 3f, g). We concluded that PAF1c is required to limit DNA damage in cells

**Fig. 2 | PAF1c maintains genomic stability in PDAC cells. a** Quantitative image-based cytometry with DNA content (Hoechst) shown on the *x*-axis. Left: EdU intensity shown on the *y*-axis (number of cells plotted: siNTC: *n* = 1032, siEXOSC5:*n* = 1029). Middle: pKAP1(S824) intensity shown on the *y*-axis (number of cells plotted: siNTC: *n* = 1048; siCDC73: *n* = 833). Right: pKAP1(S824) intensity upon incubation with AZD6738 (0.1 µM; 24 h) (number of cells plotted: siNTC: *n* = 1654; siCDC73: *n* = 1175). For each parameter, one siRNA that scored as a hit is shown as an example. Representative images are shown at the bottom. Scale bar: 10 µm. siRNA transfection was 48 h and EdU was added 30 minutes before fixation (number of independent experiments *n* = 3, except for "+ ATRi", *n* = 2). **b** Venn diagram showing the overlap between hits of different read-outs. Targets scoring in at least 2 of the 3 read-outs are shown. siRNAs with *P* < 0.05 ("+ATRi" *P* < 0.15) were considered as

hits. **c** Immunoblot of cells harboring doxycycline-inducible shRNA targeting luci-ferase, components of the PAF1c and *MYC*. Depletion was induced by doxycycline addition for 72 h and AZD6738 treatment for 72 h at 0.2 µM concentration. * refers to a non-specific band. Beta-actin was loading control (*n* = 3). **d** Growth curve of cells harboring doxycycline-inducible shRNA targeting *CTR9*. "-CTR9" indicates that shRNA expression was induced by doxycycline (1 µg/ml), "+CTR9" samples were treated with ethanol. AZD6738 (0.5 µM) was added where indicated. Each data point represents an independent experiment (*n* = 2 independent experiments). **e** Bar plots showing percentage of pKAP1-positive cells. Where indicated, doxycycline (1 µg/ml) was added for 48 h. Cells were treated with indicated compounds for 24 h. Data are presented as mean ± s.d. (*n* = 3 independent experiments; unpaired two-sided *t*-test). Source data are provided as a Source Data file.

expressing high MYC levels and that this correlates with a requirement for PAF1c in MYC multimerization.

## PAF1c promotes transcription elongation of genes that are essential in S-phase

The PAF1 complex has multiple functions in transcription-associated processes[25] including transcription elongation by RNAPII[33], facilitation of promoter-proximal double-strand break repair[22] and transcription termination[34]. To understand how PAF1c limits double-strand break accumulation during S-phase, we first used proximity ligation assays for the PAF1c subunit CTR9 and transcriptionally active RNAPII phosphorylated at serine 5, to show that endogenous MYC is required for the association of PAF1c with RNAPII (Fig. 3a). We then performed a second siRNA screen of downstream effectors and regulators of PAF1c function to decipher which of its possible functions is critical for its role in limiting DNA damage during S-phase, using phosphorylation of KAP1 in the presence of low concentrations of AZD6738 as readout (Fig. 3b). In these experiments, we used depletion of the two PAF1c subunits CDC73 and RTF1 as positive controls. Depletion of either subunit as well as depletion of HUWE1, which mediates the transfer of PAF1c from MYC to RNAPII[22], slightly increased KAP1 phosphorylation, which was enhanced upon incubation with low dose AZD6738. In contrast, depletion of factors involved in PAF1c-dependent promoter-proximal histone modifications (RNF20, RNF40, UBE2A,UBE2B and WAC) did not increase KAP1 phosphorylation nor did depletion of INO80, which removes RNAPII during TRCs in yeast[35–38]. We confirmed that depletion of RNF20 essentially abolished H2B (K120) ubiquityla-tion but had no detectable effect on phosphorylation of KAP1 either in the absence or presence of AZD6738 (Supplementary Fig. 4a). Deple-tion of several subunits of the CUL5/Elongin complex (CUL5, TCEB1, TCEB2, TCEB3)[39], which ubiquitylates stalled RNAPII, or the TIF1γ/ TRIM33 protein, which controls elongation of genes of the erythroid lineage[40], also did not increase KAP1 phosphorylation. In contrast, depletion of cyclin K and its associated kinase, CDK12, as well as CDK9, increased double-strand break formation (Fig. 3b). PAF1c recruits CDK12 for pause release and transcriptional elongation[41]. Collectively, the data argue that the canonical function of PAF1c in transcription elongation is critical in this assay.

We therefore searched for PAF1c-dependent genes that may cause these phenotypes and performed RNA sequencing after depletion of the PAF1c subunit, CTR9, comparing the results to previous results obtained after MYC depletion[7]. Immunoblots showed that the shRNA used effectively decreased levels of CTR9 (Supplementary Fig. 4b), and ChIP-sequencing assays showed that almost all CTR9 had been removed from chromatin upon doxycycline addition (Supplementary Fig. 4c). As previously observed in U2OS cells, depletion of CTR9 led to a moderate reduction in the global chromatin occupancy of endo-genous MYC (Fig. 3c)[22]. This moderate decrease in MYC chromatin binding as well as the depletion of CTR9 itself had no significant effect on the expression of canonical MYC target genes, such as those included in the hallmark gene sets of MYC targets (Fig. 3d). We

confirmed this observation by depleting a second PAF1c subunit, CDC73 (Supplementary Fig. 4d). GO term analysis confirmed that depletion of CTR9 or CDC73 had no effect on expression of canonical MYC target genes involved in, for example, ribosome biogenesis and ribosomal RNA processing (Fig. 3e). However, like MYC, depletion of CTR9 or CDC73 downregulated a set of genes that is highly enriched for genes involved in DNA replication and repair and includes genes that are critical for maintaining genomic stability during S-phase (Fig. 3e; Supplementary Fig. 5a). We confirmed these observations using RT-qPCR assays (Fig. 3f). To understand this selectivity, we per-formed 4sU-sequencing of nascent RNA after CTR9 depletion and plotted the change in occupancy relative to gene length; the results are shown as a metagene plot for quartiles of increasing gene length (Fig. 3g). This showed that CTR9 depletion caused a significant decrease in nascent transcription in the gene body of long genes and that the strength of the effect increased with gene length, consistent with the role of PAF1c as an elongation complex. At the same time, CTR9 depletion caused a global increase in nascent transcription in promoter-proximal regions (Fig. 3g). Consistent with this, CTR9 depletion preferentially upregulated the expression of short genes, but downregulated the expression of very long genes (Supplementary Fig. 5b). Importantly, many genes that are critical for DNA synthesis and repair are very long (Supplementary Fig. 5a), arguing that the decrease in their expression upon CTR9 depletion is a consequence of their length. Indeed, analysis of 4sU data from several long genes involved in S phase progression and DNA repair confirmed that their transcription terminates prematurely upon CTR9 depletion (Fig. 3h). Consistently, CTR9 depletion decreased the total levels of several key DNA repair proteins (Supplementary Fig. 5c). Bioinformatic analyses showed that CTR9 depletion preferentially downregulated the expression of long, but not of short, genes involved in DNA synthesis and repair (Supplementary Fig. 5d). We concluded that PAF1c is spe-cifically required for elongation of long genes and that its role in lim-iting double-strand break accumulation upon ATR inhibition reflects the fact that many genes required for replication and DNA repair are very long[42].

## CTR9 is required for PDAC maintenance in vivo

To determine whether PAF1c is required for PDAC growth, we gener-ated stable clones expressing a doxycycline-inducible shRNA targeting CTR9 (shCTR9) and characterized them by immunoblotting. We selected two clones and injected them into syngeneic C57BL/6 mice (Supplementary Fig. 6a). Seven days later, we determined the tumor status using luciferase imaging and added doxycycline to the food to induce CTR9 depletion. Subsequently, we monitored tumor growth and survival. As controls, we used tumors harboring shCTR9 that were kept in the absence of doxycycline as well as tumors expressing a control non-targeting shRNA in the presence of doxycycline. The analysis showed that depletion of CTR9 led to an often complete tumor regression, documented by very large decreases in luciferase activity such that many tumors were virtually undetectable by

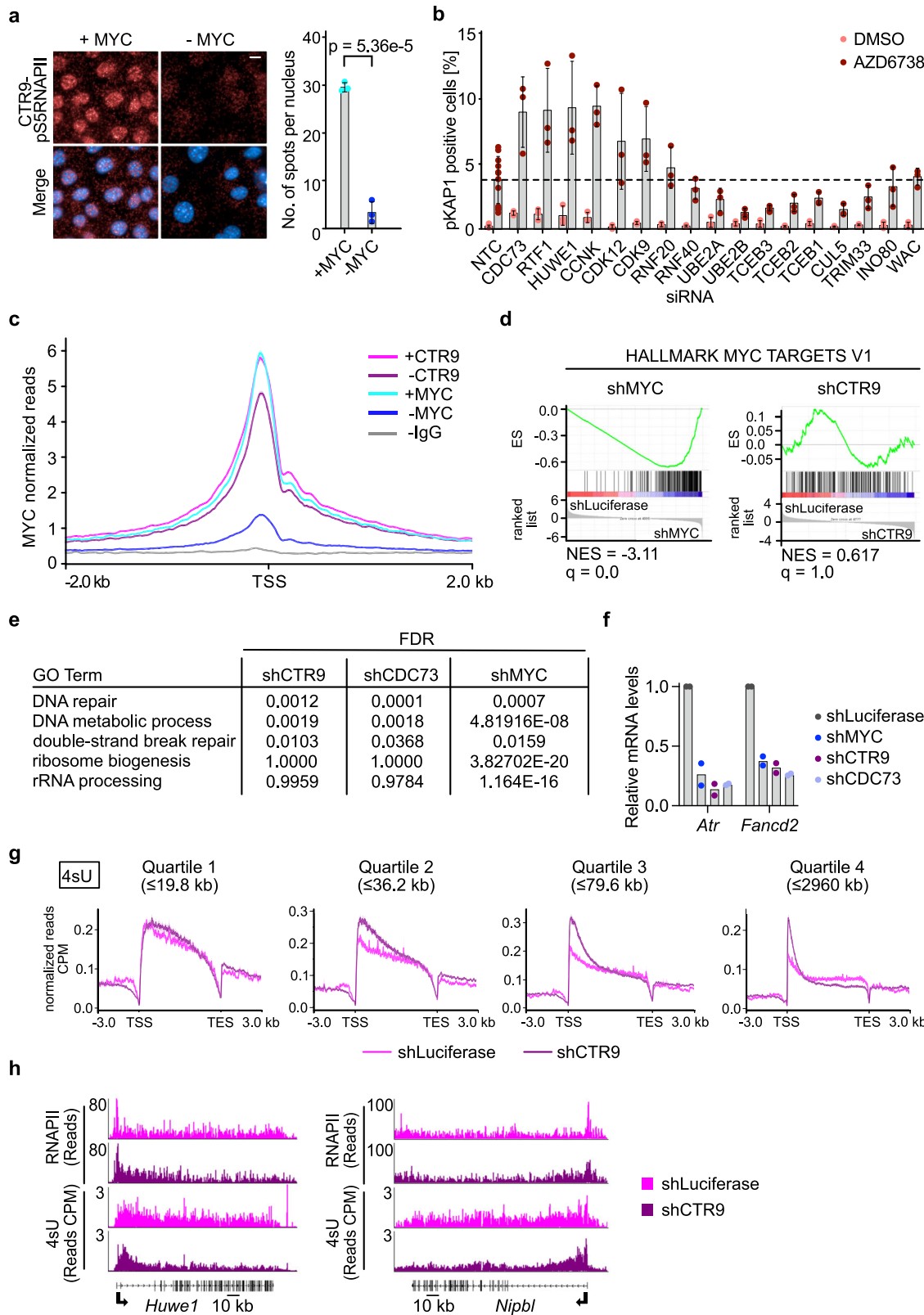

luciferase imaging after CTR9 depletion (Fig. 4a, b). Intriguingly, while all mice kept in the absence of doxycycline and control mice expressing a control shRNA died within 44 days after transplantation, a subset of mice transplanted with tumors in which CTR9 was depleted survived for more than 200 days, even though doxycycline administration was ended after 91 days (Fig. 4c). This indicates that no tumor cells

capable of re-initiating tumor growth were left after treatment. We concluded that CTR9 is required for PDAC maintenance in vivo.

To test whether this effect is a consequence of PAF1c's ability to limit S phase-associated DNA damage, we made use of the observation that the ATR inhibitor AZD6738 can be used in vivo and tested whether depletion of CTR9 and of MYC sensitizes tumors to treatment with

**Fig. 3 | Characterization of PAF1c function in transcription elongation. a** (Left) Representative images of PLAs between CTR9 and pS5RNAPII in cells harboring doxycycline-inducible shRNA targeting *MYC*. Scale bar: 10 μm. (Right) Quantification of the nuclear PLA foci. Nuclei were stained with Hoechst. Data are presented as mean ± s.d. (*n* = 3 independent experiments; unpaired two-sided *t*-test). **b** Bar plot showing percentage of pKAP1-positive nuclei. siRNAs were transfected 48 h before fixation and AZD6738 (0.1 μM) was added for 24 h. Data are presented as mean ± s.d. (*n* = 3 independent biological replicate for all siRNAs except siNTC, *n* = 12). **c** Average density plot of MYC occupancy localized around the transcription start site (TSS) of all expressed (10920) genes analyzed by CUT&RUN (*n* = 2). "-CTR9/MYC" indicates samples where CTR9 or MYC depletion was induced by doxycycline (1 μg/ml) for 48 h in cells harboring doxycycline-inducible shRNA targeting *CTR9*, and "+CTR9/MYC" ethanol-treated cells. IgG was used as negative control. **d** GSEA enrichment plot of the gene set "HALLMARK MYC TARGETS V1" in cells expressing shRNA targeting luciferase versus either *MYC* (left) or *CTR9* (right) (*n* = 3). **e** Table showing downregulated GO terms upon either CTR9, CDC73 or MYC

depletion. Enrichr package in R was used for the GO term search and *P* values were calculated using Fisher's exact test. False discovery rate (FDR) is calculated using the Benjamini–Höchberg procedure to adjust *P* values for multiple comparisons. **f** RT-qPCR measurement of *Atr* and *Fancd2* mRNA levels upon depletion of either MYC, CTR9 or CDC73 relative to the control. Doxycycline addition was done for 48 h. Each data point represents an independent biological replicate (*n* = 2 biological replicates). **g** Metagene plots mapping nascent transcription marked by 4sU incorporation. Metagene plots show read density averaged over the gene bodies of all expressed genes stratified by increasing length from quartile 1 (shortest) to quartile 4 (longest), where each quartile contains 2091 or 2092 genes. Only intronic 4sU-seq reads were considered (*n* = 3). **h** Browser picture showing read distribution of RNAPII ChIP-sequencing (top) and 4sU- sequencing showing nascent transcription (bottom) at two long DNA repair (GO-0006281) genes (*n* = 2, for the RNAPII-ChIP-sequencing. *n* = 3, for the 4sU-sequencing experiment). Source data are provided as a Source Data file.

AZD6738. Consistently, exposure of mice to AZD6738 significantly enhanced the percentage of tumor cells that stained positive for phosphorylated KAP1 and H2AX in CTR9-depleted tumors while treatment of control animals with AZD6738 did not induce a significant increase in KAP1 or H2AX phosphorylation (Fig. 4d, e). Surprisingly, however, treatment with AZD6738 did not lead to a significant increase in the percentage of mice surviving after CTR9 depletion (Fig. 4f), similar to its effect in control mice (Fig. 4f). We also tested the effect of AZD6738 on the survival of mice bearing MYC-depleted tumors and noted again a small extension of survival, but no clear sign of synergy (Fig. 4f). Collectively, the data argue that limiting S phase-associated DNA damage is not sufficient to explain the ability of PAF1c disruption to enable on long-term survival of PDAC bearing mice.

### PAF1c depletion redistributes RNAPII and SPT6 to short genes including MHC class I genes

To search for additional mechanisms that contribute to the therapeutic efficacy of CTR9 depletion, we revisited the RNA sequencing data and noted that depletion of either MYC or CTR9 caused a significant and specific increase in expression of multiple MHC class I genes (Fig. 5a and Supplementary Fig. 6b). Depletion of CDC73 confirmed that these observations were due to loss of PAF1c function (Supplementary Fig. 6b). Flow cytometry analyses confirmed that the expression of H2-D1 and H2-K1, two MHC class I antigens, increased on the cell surface after CTR9 depletion (Supplementary Fig. 6c). A side-by-side comparison showed that depletion of CTR9 elevated expression H2-D1 and H2-K1 more strongly that depletion of MYC (Supplementary Fig. 6d), despite very effective MYC depletion (Fig. 1a). Furthermore, depletion of CTR9 upregulated both genes in KPC-MYCER cells even when MYCER had been activated, demonstrating that high MYC levels do not bypass CTR9 function (Supplementary Fig. 6d). These observations caught our attention, since suppression of MHC class I gene expression is a major mechanism of MYC-mediated immune evasion[43,44] and the observation suggested that depletion of CTR9, like that of MYC, might render tumors visible for the host immune system.

To understand the mechanism underlying this regulation, we initially tested whether depletion of CTR9 activates innate immune signaling pathways, since depletion of MYC causes activation of the TBK1 kinase[7] and since multiple nucleic acid species that result from DNA damage and aberrant RNA processing can signal to the innate immune system. Surprisingly, TBK1 autophosphorylation, which reflects its activity, increased as expected in response to MYC depletion, but not in response to CTR9 depletion (Supplementary Fig. 6e). Consistently, depletion of MYC, but not of CTR9 or CDC73, caused a broad increase in the expression of genes involved in interferon signaling and of cytokines, many of which are target genes of IRFs (interferon regulatory factor) and NF-κB, downstream targets of TBK1

(Supplementary Fig. 6f)[45,46]. Depletion of CTR9 also upregulated MHC class I expression in additional p53-mutant, in p16^ink4a^-deficient murine PDAC cells as well as in human PA-TU-8988T and PANC1 PDAC cells (Supplementary Fig. 6g, h). It is also detectable in published datasets of RNAPII ChIP-sequencing experiments performed in DLD-1 colorectal adenocarcinoma cell line in which the PAF1 protein, another subunit of PAF1c, has been replaced by an auxin-degradable chimera[47] (Supplementary Fig. 6i), further supporting the notion that the PAF1c is directly involved in regulation of MHC class I genes.

To understand the underlying mechanism, we first noted that genes involved in antigen presentation are significantly shorter than genes involved in DNA repair and replication, raising the possibility that the effects of PAF1c depletion on transcription elongation are the critical factor for MHC class I upregulation (Fig. 5b). We therefore performed ChIP-sequencing for RNAPII from control and CTR9-depleted cells; this showed that - paralleling the data on nascent transcription - CTR9 depletion caused a global redistribution of RNAPII in the gene body from the 3′-part of genes to promoter-proximal regions and that the strength of the effect increased with gene length (Supplementary Fig. 7a). Annotation of individual genes showed MHC class I genes are among the short genes at which RNAPII increases throughout the entire gene body after CTR9 depletion (Supplementary Fig. 7a). Inspection of browser tracks of two major MHC class I genes showed that RNAPII was loaded on the promoter of these genes in both control and CTR9-depleted cells, and RNAPII was released into productive elongation upon CTR9 depletion (Fig. 5c). We confirmed this using ChIP-qPCR of the promoter and gene body of four genes involved in antigen presentation (Fig. 5d). Next, we analyzed chromatin association of several elongation factors to understand which might be limiting for antigen presentation genes due to their being sequestered in the body of long genes. ChIP-sequencing showed that SPT5, one of two components of the DSIF (DRB sensitivity inducing factor) was loaded on RNAPII at the start site of MHC class I genes in the presence and was, like RNAPII, released into the gene body upon CTR9 depletion (Fig. 5c). Similarly, the effects of CTR9 depletion on phosphorylation of Ser2 and Ser5 of the CTD of RNAPII and on chromatin association of the SPT4 subunit of DSIF were consistent with the effects on total RNAPII and argued that RNAPII is actively elongating on MHC class I genes after CTR9 depletion (Fig. 5d). Furthermore, depletion of CTR9 decreased chromatin association of another subunit of PAF1c, RTF1, which directly promotes elongation of RNAPII (Fig. 5d)[48,49]. In contrast, depletion of CTR9 caused a large increase in association of SPT6 at both the promoter and body of MHC class I genes (Fig. 5c, d). SPT6 is a histone chaperone that mediates the displacement and reassembly of nucleosomes during transcription elongation and stimulates RNAPII processivity[50,51]. Genome-wide analyses confirmed that depletion of CTR9 led to a strong decrease of SPT6 occupancy in the body of long genes, and an accumulation on

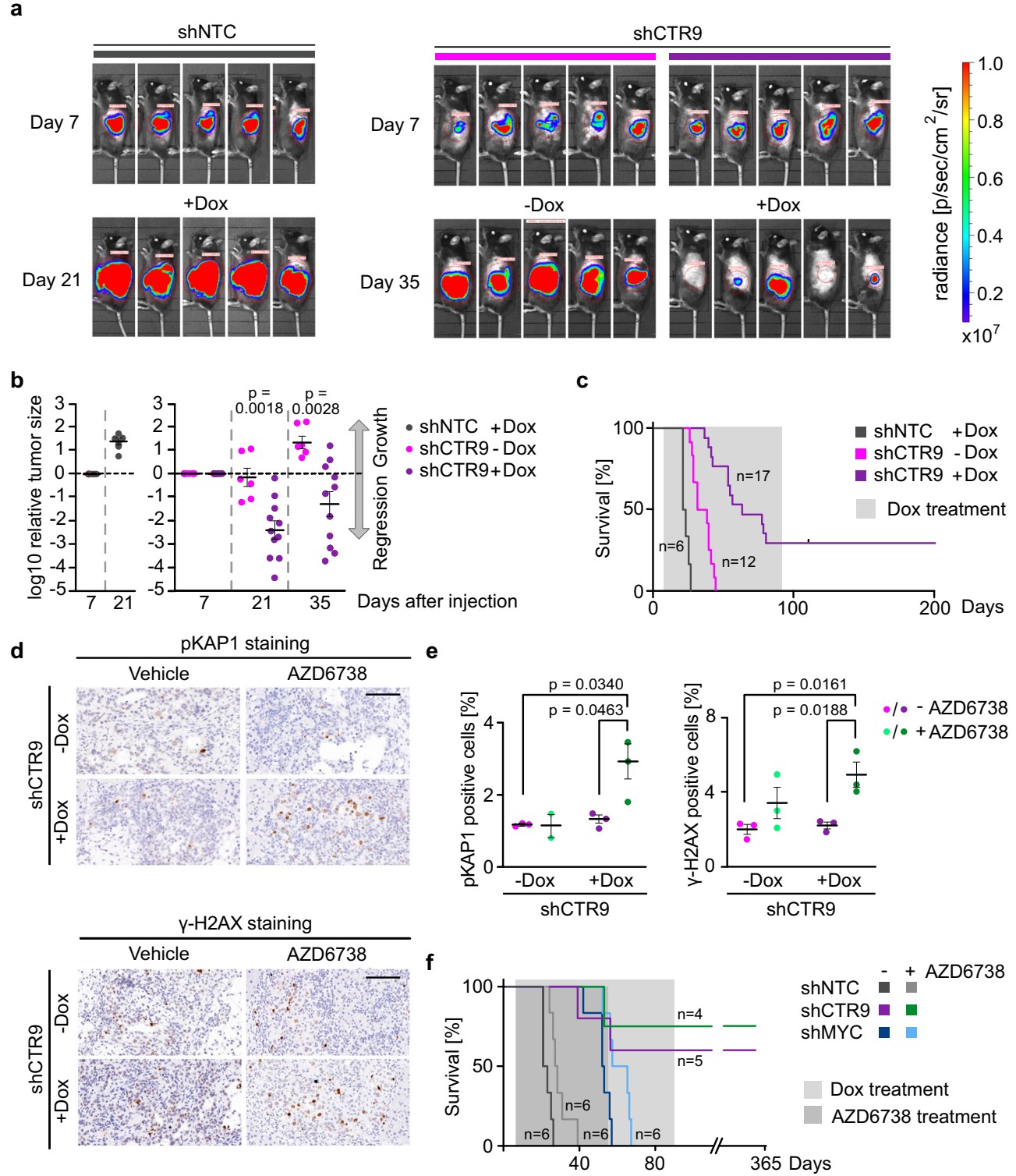

short genes and on promoter-proximal regions (Fig. 5e). Comparing the change in RNAPII, SPT5 and SPT6 occupancy for length-dependent quartiles of genes showed that the length-dependent decrease in SPT6 closely paralleled the decrease in nascent transcription (compare Fig. 3g and Supplementary Fig. 7b). This suggested that sequestration of SPT6 is one - of potentially several - mechanisms by which elongating RNAPII complexes on long genes suppress expression of MHC class I genes. Consistent with this hypothesis, depletion of SPT6 abrogated upregulation of two MHC class I genes that we tested upon CTR9 depletion (Fig. 5f). Collectively, our data show that PAF1c

promotes elongation on long genes that include DNA replication and repair genes and that the resulting sequestration of the SPT6 histone chaperone and potentially additional elongation factors limits RNAPII elongation of short genes that include genes involved in antigen presentation.

## CTR9 is critical for immune evasion of PDAC

To test whether escape from immune surveillance is the critical function of PAF1c in KPC tumors, we performed histological analyses and found that depletion of CTR9 caused an increase in the number of

**Fig. 4 | CTR9 is required for PDAC growth. a** Luciferase imaging of KPC-cell–derived tumors expressing NTC (non-targeting control) shRNA or shRNA against *CTR9* upon orthotopic transplantation into C57BL/6 J mice. Doxycycline treatment was started 7 days after transplantation. Luciferase activity is shown at day 7 after transplantation and after 2 or 4 weeks of doxycycline treatment. **b** Change of tumor size of KPC tumors expressing doxycycline-inducible shNTC (left) or shCTR9 (right) relative to the start of doxycycline treatment on day 7. The plot shows the $\log_{10}$ of the ratio of luciferase activity at the indicated times relative to day 7. Results are presented as mean ± S.E.M. *P* values were calculated using the unpaired two-sided *t*-test. *n* = 6 mice for shNTC+Dox and shCTR9-Dox, while *n* = 11 mice for shCTR9+Dox. **c** Kaplan−Meier plot of mice transplanted with KPC cells expressing doxycycline-inducible shCTR9 or shNTC. Doxycycline was added to the food for 12 weeks as indicated. **d** Immunohistochemistry of pancreatic tumors

expressing shRNA against *CTR9*. Mice were treated for 3 days with or without doxycycline and AZD6738 or vehicle. Sections were stained with anti-pKAP1 and anti-γ-H2AX antibodies. Scale bar: 100 μm. **e** Percentage of pKAP1 or γ-H2AX positive cells present in the tumor tissue. Positive cells were counted after 3 days of CTR9 depletion and AZD6738 treatment. Results are presented as mean ± S.E.M. Each dot represents an independent tumor and *P* values were calculated using the unpaired two-sided *t*-test. *n* = 3 independent tumors except for pKAP1 staining of AZD6738 treated mice, *n* = 2. **f** Kaplan−Meier plot of mice transplanted with KPC cells expressing doxycycline-inducible shCTR9, shMYC or shNTC. Expression of shRNAs was induced with doxycycline from day 7 for 12 weeks in all mice. Mice were treated with AZD6738 or vehicle from day 7 for 7 weeks. Source data are provided as a Source Data file.

multiple types of immune cells in the tumor tissue (Fig. 6a; Supplementary Fig. 8a). To better understand the significance of these changes, we isolated live cells from tumor tissue and comprehensively profiled changes in immune cell populations by measuring established surface antigens using flow cytometry (Fig. 6b; Supplementary Fig. 8b). Since only 3−5% of live cells in these isolates were tumor cells as judged by EpCAM staining, these analyses detect relative changes in immune cell populations in the tumor. In these experiments, we did not detect any significant changes in the number of B cells, the polarization of macrophages or in multiple populations of myeloid cells (Supplementary Fig. 8c). In contrast, depletion of CTR9 caused an increase in the number of (CD3-positive) T-cells, and of cDC1 cells, which are dendritic cells that stimulate T-cell responses against tumors and locally present tumor antigens to re-stimulate T-cells (Fig. 6b)[52]. Previous work has shown that there is a specific increase in expression of CTLA4, a negative regulator of T-cell receptor signaling, during tumor progression in a KRAS-driven murine PDAC model and that induction of MYC in a model of hepatocellular carcinoma elevates CTLA4 expression on T-cells[53,54]. Consistent with these observations, T-cells in control mice expressed high levels of CTLA4 and there was a robust decrease in CTLA4 expression upon CTR9 depletion in CD3- and CD8-positive T cells (Fig. 6b, c). In contrast, we did not observe changes in PD-1 expression in T-cells (Fig. 6b). The data show that depletion of CTR9 in the tumor tissue causes the loss of signals that restrain T-cell function and induce a specific activation of T-cell mediated immune responses. In strong support of this hypothesis, a large percentage of the tumor cells that remained after three days of doxycycline addition were in direct association with CD8-positive cytotoxic T-cells (Fig. 6d). To test whether restoration of immune surveillance is the critical tumor-suppressive effect of CTR9 depletion in PDAC tumors, we transplanted the KPC cells expressing inducible shCTR9 described above into NRG mice, which lack both T- and B-lymphocytes as well as natural killer cells. Under these conditions, CTR9 was completely dispensable for tumor growth since depletion of CTR9 had no significant effect on either tumor size (Fig. 6e) or survival (Fig. 6f). We concluded that the critical function of CTR9 in pancreatic carcinoma is to protect tumor from eradication by the immune system. A model summarizing our findings is shown in Fig. 7.

## Discussion

Enhanced and deregulated expression of MYC has widespread effects on cell growth, metabolism, and cell proliferation, and many of these effects are mediated by binding of MYC to E-box sequences in promoters and subsequent regulation of downstream target genes[11]. This has led to the concept that oncogenic effects of MYC are mediated by the altered expression of a large set of target genes and that strategies to target MYC must block E-box dependent activation. However, many potentially oncogenic functions of MYC overlap with those of other oncogenes: for example, the effects of MYC on cell proliferation are similar to those of the RAS pathway with respect to induction of cyclin genes[55,56] and to p53 loss with respect to repression of CDKN1A

(P21CIP1) expression[57–59]. Similarly, the ability of MYC to promote the biosynthesis of metabolic precursors[60] may be largely alleviated by KRAS-dependent macropinocytosis[61]. Consequently, the critical oncogenic functions of endogenous MYC in the presence of the KRAS and p53 mutations that drive the development of most human PDACs[1] are less clear. Endogenous MYC is required in PDAC for progression through the S-phase of the cell cycle and for immune evasion of tumors[6,7,62]. At the same time, biochemical analyses show that the MYC protein participates in several different complexes, raising the question of which complexes are required to carry out both functions.

Here, we performed a focused siRNA screen searching for proteins that limit DNA damage during S phase of PDAC cells. This screen identified the RNAPII-associated complex PAF1c: while depletion of CTR9 or CDC73 had no significant effect on the occurrence of TRCs, depletion of several PAF1c subunits strongly enhanced the accumulation of double-strand breaks upon ATR inhibition arguing that PAF1c limits the DNA damage arising during S-phase. Activation of MYCER enhanced the DNA damage associated with CTR9 depletion and the dependence on multiple PAF1c subunits closely correlated with high MYC levels and high MYC target gene expression across multiple human cell lines, arguing that PDAC cells with high MYC levels are particularly dependent on PAF1c. At the same time, MYC binds directly to PAF1c and activation of MYC in osteosarcoma cells recruits PAF1c to its target promoters and, via PAF1c, promotes double-strand break repair at promoters[63]; we show here hat endogenous MYC in pancreatic carcinoma cells is required to recruit PAF1c to RNAPII. Collectively, the data reveal a PAF1-dependent genome protective function of MYC that is distinct from its regulation of canonical target genes.

A secondary siRNA screen of downstream mediators of PAF1c showed that BRE1 does not mediate the function of MYC and PAF1c in suppressing DNA damage during S-phase in KPC cells. Instead, the screen identified the HUWE1 ligase that transfers PAF1c from MYC to RNAPII[22], and the transcription-related kinases CDK9 and CDK12 as critical for DNA damage upon ATR inhibition. This suggests that the canonical function of the PAF1c in RNAPII elongation[33] is critical in KPC cells. Consistently, depletion of CTR9 decreased nascent transcription and RNAPII, SPT5 and SPT6 occupancy in the body of multiple long genes required for DNA repair and S-phase progression and downregulates these genes. Most likely, therefore, PAF1c depletion causes a critical decrease in the levels of proteins required to prevent accumulation of double-strand breaks during S-phase.

Depletion of the PAF1c subunit CTR9 in a PDAC transplant model demonstrated its critical role in the growth and maintenance of PDACs in vivo since many tumors undergo complete regression upon CTR9 depletion. Strikingly and in contrast to depletion of MYC, a subset of mice shows long-term survival even after doxycycline withdrawal, demonstrating that no tumor cells capable of initiating tumor regrowth remain. We have previously found that PDAC tumors that relapse after MYC depletion restore MYC function in part by downregulating the expression of the MXD family of MYC antagonists, arguing that, relative to MYC, CTR9 function is tumorigenesis is less

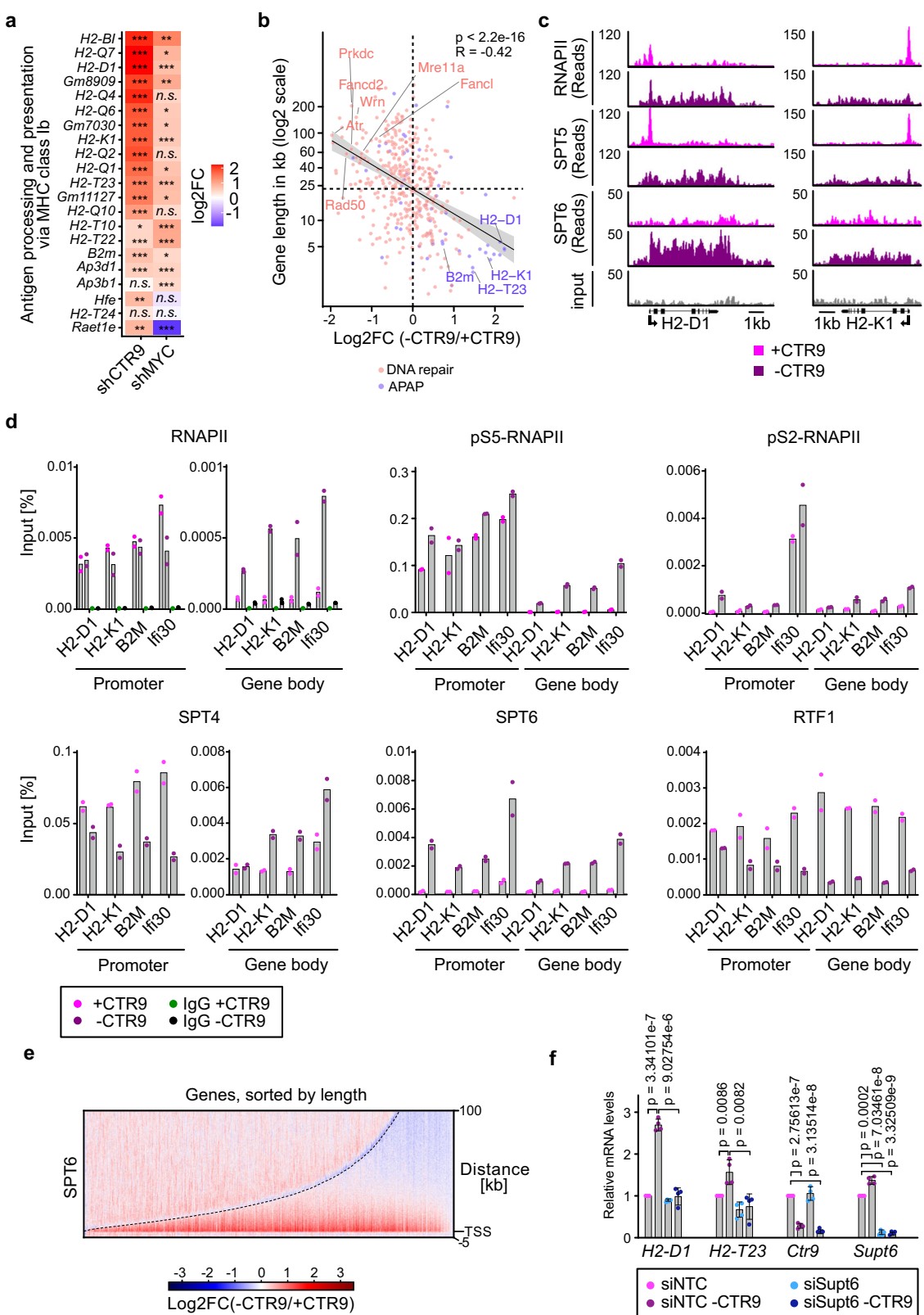

easily bypassed by compensatory mechanisms[7]. CTR9 depletion in combination with ATR inhibition induces double-strand break formation in culture and in vivo, arguing that CTR9 prevents S phase-associated DNA damage. Surprisingly, while CTR9 depletion showed synergy with ATR inhibition in inducing DNA damage, there was no synergistic effect on survival. Instead, the survival benefit conferred by CTR9 depletion was entirely dependent on the host immune system

and correlated with elevated mRNA levels of multiple MHC class I molecules. Repression of MHC class I genes by MYC[44] and MYCN[43] is consistently observed in multiple experimental models and indeed many MYC-driven tumors are immunologically "cold" tumors. Reflecting these tumor cell-intrinsic changes, depletion of CTR9 induced a specific increase in the number of T-cells and cDC1 dendritic cells, which present antigen to T-cells, and an activation of CD8-

**Fig. 5 | PAF1c sequesters SPT6 on long genes. a** Heat map showing log₂FC of change in expression of genes involved in antigen processing and presentation via MHC class I molecules (GO-0002475) upon the depletion of either CTR9 or MYC ($n = 3$). *P* values were calculated using the likelihood ratio test (LRT), then False discovery rate (FDR) was calculated using the Benjamini–Höchberg procedure to adjust *P* values for multiple comparisons. The FDRs were: **p ≤ 0.05, **p ≤ 0.001, ***p ≤ 0.0001, n.s. not significant. **b** Scatter plot of gene length versus log₂ fold-changes in gene expression upon CTR9 depletion, comparing expressed genes of the GO term DNA repair (GO-0006281, $n = 375$ genes) in red with antigen processing and presentation genes (APAP, GO-0019882, $n = 56$ genes) in blue (Pearson correlation coefficient (R) = −0.42, $p < 2.2e$-16, two-sided pearson correlation test). **c** Browser picture showing read distribution of RNAPII (top), SPT5 (middle) and SPT6 (bottom) at *H2-K1* and *H2-D1* genes from a ChIP-Rx experiment in cells expressing doxycycline-inducible shRNA targeting *CTR9* ($n = 2$). "-CTR9" indicates

that shRNA expression was induced by doxycycline (1 μg/ml), "+CTR9" samples were treated with ethanol. **d** ChIP-qPCR showing RNAPII, pS5RNAPII, pS2RNAPII, SPT4, SPT6 or RTF1 binding to the indicated genes either at the promoter or in the gene body. IgG was used as a negative control to show the background signal. Data are presented as mean of two technical replicates of one of two biological replicates with similar results ($n = 2$). **e** Heat map of the log₂FC of SPT6 occupancy for all expressed (10,920) genes analyzed by ChIP-Rx comparing CTR9 depletion to the control. Genes are sorted according to their length from shortest (left) to longest (right). The black dashed line represents the end of the gene ($n = 2$). **f** RT-qPCR measurement of *H2-D1* and *H2-T23* MHC class I genes, *CTR9* and *Supt6* (encoding SPT6). KPC cells expressing doxycycline inducible shRNA targeting *CTR9* were used. Both siRNA transfection and doxycycline addition were for 48 h. Data are presented as mean ± s.d. ($n = 4$ independent experiments; unpaired two-sided *t*-test). Source data are provided as a Source Data file.

positive T-cells. On a molecular level, this correlated with a strong decrease in the expression of the checkpoint protein CTLA-4, which inhibits T-cell receptor signaling; previous work had shown that induction of MYC in tumor cells induces CTLA4 expression in T-cells in a model of hepatocellular carcinoma, and that CTLA4 increases during tumor progression a KRAS-driven PDAC model[53,54].

Various types of nucleic acids such as single-stranded DNA, double-stranded RNA and R-loops, resulting from DNA damage and abnormal transcription, can be exported from the nucleus and activate innate immune signals and the TBK1 kinase via pattern recognition receptors located in the cytosol or endosomes[64,65]. Surprisingly, depletion of PAF1c subunits resulted in specific upregulation of MHC class I molecules in the absence of TBK1 activation, suggesting that there is a direct link between transcriptional elongation and immune evasion. Subsequent analyses showed that depletion of CTR9 releases RNAPII, SPT5 and, most clearly, SPT6 from the body of long genes, causing a global redistribution of the transcription machinery towards promoter-proximal regions. This correlates with an upregulation of expression of short genes, which includes MHC class I genes. Previous work has shown that MYC has widespread effects on multiple steps of RNAPII assembly, with effects on pause release and elongation predominating in several systems[12,66]. Our data suggest that MYC-dependent immune evasion is a direct consequence of its effects on transcriptional elongation. While several molecular mechanisms can account for these observations, we propose that long and short genes compete for components of the basal transcriptional machinery and that this competition is a central mechanism of MYC-dependent immune evasion. Since PAF1c, unlike most of the MYC protein, has a stably folded structure, our data suggest that small molecules targeting PAF1c may be found that specifically disrupt the immune evasive function of MYC while sparing its canonical functions in physiological cell growth.

## Methods
### Mouse experiments
All animal procedures were approved by the Regierung von Unterfranken under protocol numbers RUF 55.2–2532–2-1419 and RUF 55.2-2532-2-148. For orthotopic transplantation, mice (C57BL/6 J or NOD.Cg-*Rag1*^tm1Mom *Il2rg*^tm1Wjl/SzJ (NRG), RRID:IMSR_JAX:007799) were anaesthetized, the spleen and the pancreas were externalized and 50,000 modified KPC cells were injected in 50 μL Matrigel/PBS (2:1) into the pancreatic tail. The mice were treated prophylactically with painkillers for three days. Luciferase activity was measured with an IVIS camera, first on day 7 after orthotopic transplantation. Doxycycline was administered to the mice in the diet, starting on day 7 after orthotopic transplantation. The administration of doxycycline was stopped after 12 weeks. The inhibitor AZD6738 (MCE) was dissolved in 10% DMSO, 40% propylene glycol and 50% water. The inhibitor was administered to mice orally by gavage once a day. After every 5 days, the administration was suspended for 2 days. The first dose was given

on day 7 after orthotopic transplantation. In total, AZD6738 was administered over a period of 7 weeks (35 single doses). C57BL/6 J mice were ordered from Charles River, bred in-house, and 5–12 weeks old at the start of the experiment. NRG mice were ordered from Janvier-Labs and were 9 weeks old at the start of the experiment. Since KPC cells were established from a tumor of a male mouse, orthotopic transplantation experiments were performed using male mice to minimize the risk of immune rejection. Maximum tumor size/ burden was not exceeded. The health status of the mice was checked at least once a day. The general condition, the behavior of the animals, the body condition index, the visibility of the tumor and, in the NRG animals, the impairment due to the immunodeficiency status of the mice were evaluated. The overall condition was recorded using a scoring system, and the mice were sacrificed if there was an average burden over two days or if the average burden was exceeded.

### Flow cytometry analysis
Extracted tumors were mechanically disrupted and digested in a 500 μL mixture of 1 mg/mL collagenase A and D (Roche, #10103578001, #11088858001), trypsin inhibitor from glycine max (Sigma, #T6522) and 0.4 mg/mL DNase I (Roche, #10104159001) in PBS (Sigma, #D8537) first at 22 °C for 45 min and then at 37 °C for 15 min with 1000 rpm rotation in a thermo-mixer (Eppendorf). EDTA (Merck, #324504) was then added to a final concentration of 10 mmol/L, whereafter the cells were passed through a 70-μm mesh prior to immunostaining. Cells were stained with fixable viability dye eFluor780 (1:1000, eBioscience, #65-2860-40) for 15 min at 4 °C, blocked with anti-mouse TruStain FcX™ (1:50, Biolegend, #101319) for 5 min at 4 °C and stained for 30 min at 4 °C with combinations of the following fluorescently conjugated antibodies: anti-CD45 BV510 (30-F11, 1:300), anti-CD3e AF700 (500A2, 1:300), anti-PD-1/CD279 BV421 (29 F.1A12, 1:200), anti-CD11b PE (M1/70, 1:300), anti-CD11c AF647 (N418, 1:300), anti-CD4 BV650 or BV605 (GK1.5, 1:200), anti-CD8 FITC or PE (53-6.7, 1:200), anti-CTLA-4 PE (UC10-4B9, 1:200), anti-CD45R/ B220 PE-Dazzle (RA3-6B2, 1:200), anti-Ly6G PerCP-Cy5.5 (1A8, 1:300), anti-Ly6C AF700 (HK1.4, 1:300), anti-F4/80 BV421 (BM8, 1:300), anti-Podoplanin APC (8.1.1, 1:300), anti-Ep-CAM BV421 (G8.8, 1:300), anti-E-Cadherin PE-Dazzle (DECMA-1, 1:200), anti-CTLA-4 PerCP-Cy5.5 (UC10-4B9, 1:200) all from BioLegend and CD31 PE-Cy7 (MEC13.3, 1:200) and CD44 SB645 (IM7, 1:200) both from eBioscience and anti-MHCII SB600 (M5/114.15.2, 1:200) and anti-PDGFRa SB702 (APA5, 1:200) both from ThermoFisher. Anti-CD206 BV650 (C068C2, 1:200) was added after surface staining was completed and after fixation-permeabilization using the FoxP3/Transcription Factor Fixation/Permeabilization Kit (eBioscience, #00–5523–00) and the manufacturer's guidelines were followed. All flow cytometry was performed on an Attune NxT (Thermo Fisher) analyzer and offline analysis was performed with FlowJo software (Treestar, version 10.9).

For staining of MHC Class I proteins on the surface of the cells shown in Supplementary Fig. 6c, cells were trypsinized and washed

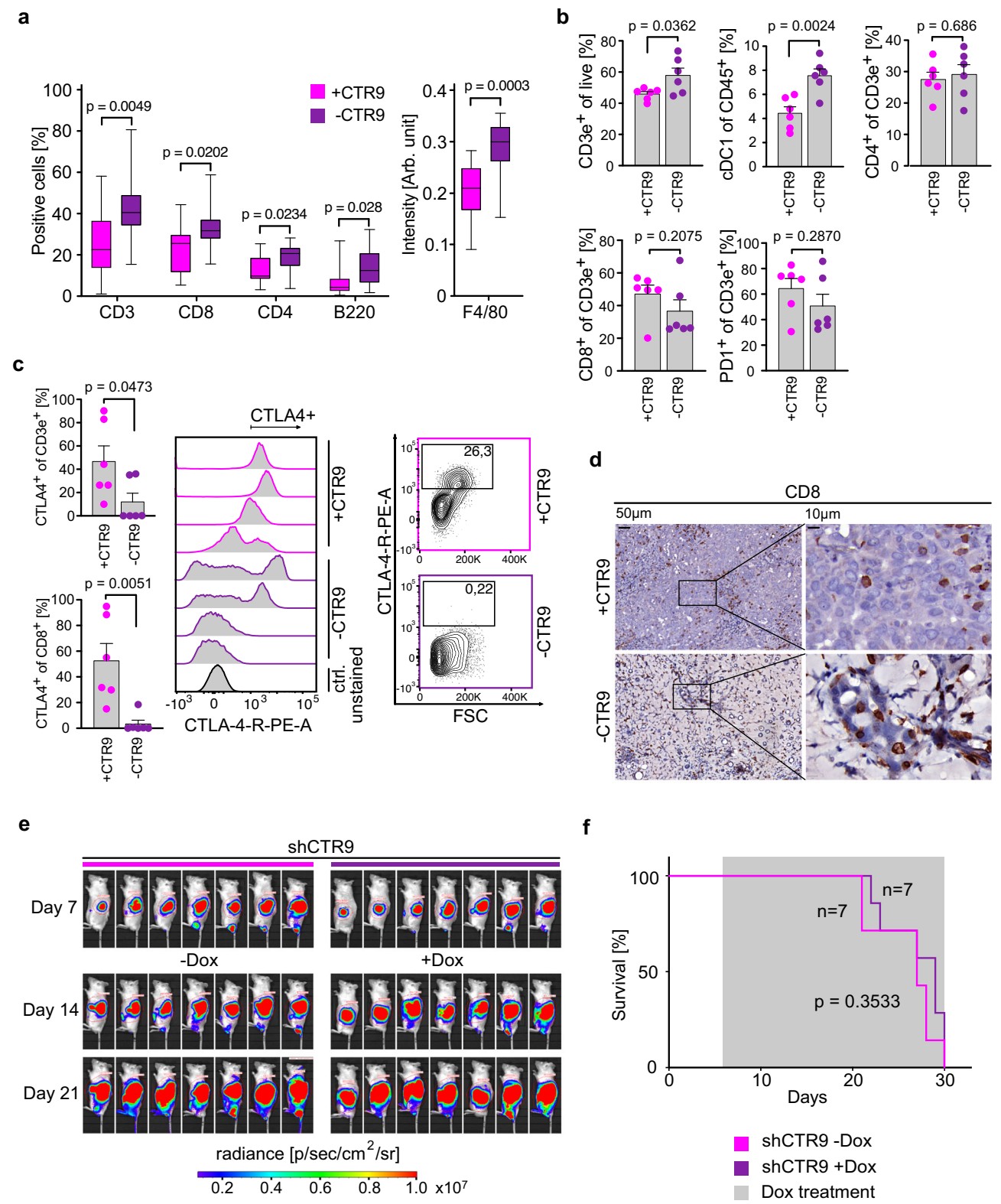

twice with PBS. Cells were blocked using 2% FCS in PBS and 1 μg first antibody was added for 30 min at 4 °C per condition. Cells were washed twice using 2% FCS in PBS and secondary antibody was added in a 1:400 dilution for 30 min at room temperature. Cells were washed twice with 2% FCS in PBS. Measurement was performed using BD FACSCanto II flow cytometry and BD FACSDIVA software. The following primary antibodies were used: anti-H-2Db (Thermo Fisher Scientific

Cat# MA5-17992, RRID:AB_2539376), anti-H-2Kb (Bio X Cell Cat# BE0172, RRID:AB_10949300).

For Annexin V/propidium iodide flow cytometry shown in Supplementary Fig. 3c, the supernatant of the respective condition was combined with the cells that were harvested by trypsinization and washed with cold PBS. The cell pellet was resuspended in 100 μL of 1× Annexin V binding buffer (10 mM HEPES (pH 7.4), 140 mM NaCl,

**Fig. 6 | CTR9 is required for immune evasion in PDAC. a** Box plots showing the percentage of the indicated cells (left), or intensity of DAB (diaminobenzidine) staining (right) in tumor tissue. Mice carrying tumors expressing doxycycline-inducible shRNA against *Ctr9* were treated for three days with doxycycline or control. For each plot, 4–6 regions per tumor were evaluated from three independent tumors, *n* = 14 regions analyzed. *P* values were calculated using unpaired two-tailed *t*-test using Welch's correction. In the box plot, the central line shows the median, the box borders extend from the 25th to the 75th percentile, and the whiskers go down to the smallest value and up to the largest. **b** Bar plots showing flow cytometry analysis of immune cell subsets in the tissues of tumors initiated by orthotopic transplantation of KPC cells harboring doxycycline-inducible shNTC or shCTR9. After 7 days of transplantation, doxycycline treatment was added for 3 days. Pre-gating was performed using single living cells. cDC1 cells were defined as (CD45$^+$, CD11b$^-$, CD11c$^+$). Data are presented as mean ± s.d. (*n* = 6 independent tumors; unpaired two-sided *t*-test). **c** Left, bar plots showing the percentage of

CTLA4$^+$ T-cells of CD3e$^+$ or CD8$^+$ cells, data are presented as mean ± s.d. (*n* = 6 independent tumors; unpaired two-sided *t*-test). Middle, histograms showing staining of CTLA4 on T-cells (CD3e$^+$), with intensity on the x-axis, and counts on the y-axis. Right, flow cytometry contour plot, documenting the gating strategy for CTLA4$^+$ T-cells (CD3e$^+$). **d** Representative immunohistochemistry images of tumor sections stained for CD8 (brown) with two different magnifications. Note the juxtaposition of CD8-positive T-cells with the remaining tumor cells. "-CTR9" refers to three days of doxycycline treatment. **e** Luciferase imaging of KPC cell derived tumors harboring shRNA against *Ctr9* upon orthotopic transplantation into NRG mice. Doxycycline treatment was started 7 days after transplantation. Luciferase activity of KPC cells was measured at day 7 after transplantation (start) and after 1 and 2 weeks of doxycycline treatment. **f** Kaplan–Meier plot of NRG mice transplanted with KPC cells expressing doxycycline-inducible shCTR9. shRNA was induced with doxycycline from day 7. *P* values were calculated using two-sided Mantel–Cox test. Source data are provided as a Source Data file.

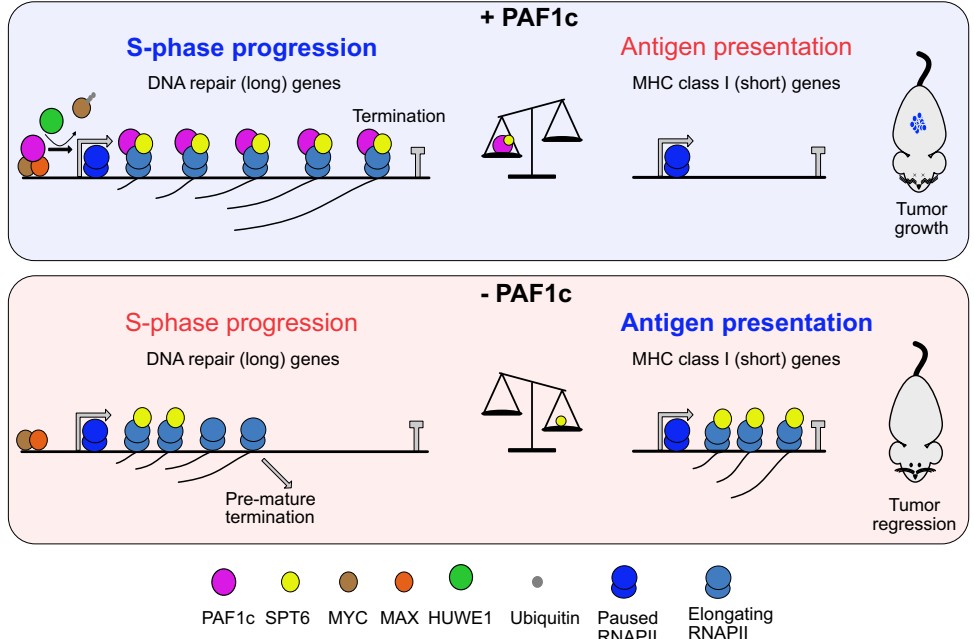

**Fig. 7 | Model summarizing our findings.** Model summarizing our findings. We propose that PAF1c acts downstream of MYC to enhance full-length transcription of long DNA repair genes while limiting the expression of short MHC class I genes. Interference with PAF1c function results in the redistribution of transcription elongation factors, including SPT6. This restores expression of short MHC class I genes and impairs the full-length transcription of long DNA repair genes. This change reshapes the tumor microenvironment, causing immune cell-dependent tumor regression and enabling long-term survival of PDAC-bearing mice.

and 2.5 mM CaCl2) with Annexin V/Pacific Blue dye. The cells were incubated for 15 min at room temperature in the dark. Subsequently, 400 μL of 1× binding buffer containing propidium iodide (54 μM) was added. The samples were stored in the cold and dark until analysis. Measurement was performed using BD FACSCanto II flow cytometry and BD FACSDIVA software.

## Immunohistochemistry
For immunohistochemical analysis, samples were embedded in paraffin and sectioned at 6 μm using a microtome (Leica). Sections were deparaffinized, rehydrated and subjected to high-temperature antigen retrieval at pH 6 for all stains except for pKAP1 at pH 9. Detection of primary antibodies was performed using peroxidase-coupled appropriate secondary antibodies. Sections were counterstained with hematoxylin. Staining was imaged using a Pannoramic DESK scanner and analysed using QuPath[67]. Due to the shape of macrophages (Supplementary Fig. 6), proper cell detection was not possible and mean intensity was used for the analysis. For histology, the following antibodies were used: anti-CD3 (Abcam Cat# ab16669,

RRID:AB_443425, 1:100), anti-CD4 (Thermo Fisher Scientific Cat# 14-9766-82, RRID:AB_2573008, 1:100), anti-CD8 (Lab Vision Cat# RB-9009-P1, RRID:AB_149750, 1:100), anti-F4/80 (Abcam Cat# ab6640, RRID:AB_1140040, 1:1,000), anti-CD45R/B220 (BD Biosciences Cat# 550286, RRID:AB_393581, 1:50), anti-KAP1 phospho-S824 (Abcam Cat# ab70369, RRID:AB_1209417, 1:500), anti-γ-H2AX (Abcam Cat# ab2893, RRID:AB_303388, 1:1,000).

## Cell culture
HEK293TN (RRID: CVCL_UL49), PA-TU-8988T, Panc-1, U2OS (RRID: CVCL_0042), and KPC cells were cultured in DMEM (Sigma-Aldrich) supplemented with 10% FCS (Biochrom and Sigma-Aldrich) and 1% penicillin-streptomycin (Sigma-Aldrich). KPC cells were provided by Jens Sieveke[7,68]. P53-mutant and p16ink4a(Cdkn2a)-deficient murine PDAC cell lines were provided by Dieter Saur. Panc-1 and PA-TU-8988T human PDAC cell lines were provided by Mathias Rosenfeldt. HEK293TN and U2OS cells were obtained from ATCC. Cells were regularly tested negative for mycoplasma contamination using standard methods (Arbeitsgemeinschaft Gentechnik, AM029).

## Live-cell imaging

KPC cells harboring shRNA targeting *CTR9* were seeded in a 24-well plate (Greiner), after the indicated treatments, cellular proliferation was assessed using the Incucyte SX5, with 9 fields analyzed every 8 h using the 10x objective.

## Chemicals

AZD6738 (ATRi) was purchased from MedChemExpress. For other compounds shown in Fig. 2e, suppliers and respective concentrations used are shown in Supplementary Data 2.

## Lentiviral infection and siRNA transfection

psPAX2 (Addgene 12260) and pMD2.G (Addgene 12259) lentiviral packaging plasmids were used to generate stable cell lines. HEK293 cells were used for lentiviral production. Cell-free virus-containing supernatants were used for infections. shRNAs against CTR9, CDC73 and RTF1 were selected as previously described[69] and lentivirally transduced into the cell genome. KPC cells were infected with lentiviral supernatants in the presence of 4 µg/ml polybrene for 24 h. Cells were selected with puromycin 2 µg/ml for 2 days. KPC cells harboring shMYC were used as described before[7]. siRNA transfection was performed using RNAiMAX reagent (Thermo Fisher Scientific) according to the manufacturer's protocol following the reverse transfection method.

## Immunoblotting

Cells were lysed using RIPA buffer (50 mM HEPES, 140 mM NaCl, 1 mM EDTA; 1% Triton X-100, 0.1% sodium deoxycholate and 0.1% SDS) containing protease and phosphatase inhibitor cocktails (Sigma-Aldrich). Lysates were cleared by centrifugation and bicinchoninic acid assay was used to determine protein concentration. Protein samples were separated on Bis-Tris gels and transferred to a polyvinylidene fluoride membrane (Millipore). Protein expression was analyzed by immunoblotting using the indicated primary antibodies. Membranes were scanned and analyzed using LAS 4000 mini-imaging system (Fuji). Vinculin or beta-actin was used as a loading control. The following primary antibodies were used: anti-MYC (Abcam Cat# ab32072, RRID:AB_731658, 1:5000), anti-CTR9 (Bethyl Cat# A301-395A, RRID:AB_960973, 1:1000), anti-CDC73 (Cell Signaling Technology Cat# 3644, RRID:AB_2078386, 1:1000), anti-RTF1 (Bethyl Cat# A300-179A, RRID:AB_2185963, 1:1000), anti-vinculin (Sigma-Aldrich Cat# V9131, RRID:AB_477629, 1:5000), anti-actin-beta (Sigma-Aldrich Cat# A5441, RRID:AB_476744, 1:5000), anti-KAP1 phospho-S824 (Abcam Cat# ab70369, RRID:AB_1209417, 1:5000), anti-KAP1 (Bethyl Cat# A300-274A (also A300-274A-M, A300-274A-T), RRID:AB_185559, 1:1000), anti-TBK1 phospho-S172 (Cell Signaling Technology Cat# 5483, RRID:AB_10693472, 1:1000), anti-TBK1 (Cell Signaling Technology Cat# 3504, RRID:AB_2255663, 1:1000), anti-NIPBL (Santa Cruz Biotechnology Cat# sc-374625, RRID:AB_10989775, 1:1000), anti-MRE11 (Abcam Cat# ab208020, RRID:AB_2814655, 1:1000), anti-CHK1 (phosphoSer345) (Cell Signaling Technology Cat# 2348, RRID:AB_331212, 1:1000), anti-RAD50 (Abcam Cat# ab208019, 1:1000), anti-ATR (Thermo Fisher Scientific Cat# A300-138A, RRID:AB_2063318, 1:1000), anti-CHK1 (Santa Cruz Biotechnology Cat# sc-7898, RRID:AB_2229488, 1:1000), anti-FANCD2 (Abcam Cat# ab108928, RRID:AB_10862535, 1:1000), anti-HUWE1 (Abcam Cat# ab70161, RRID:AB_1209511, 1:1000), anti-RNF20 (Cell Signaling Technology Cat# 11974, RRID:AB_2797786, 1:1000), anti-Ubiquityl-Histone H2B (Lys120) (Cell Signaling Technology Cat# 5546, RRID:AB_10693452, 1:1000), anti-histone H2B (Abcam Cat# ab1790, RRID:AB_302612, 1:1000). Depending on the primary antibody, either ECL anti-mouse IgG horseradish peroxidase (GENA931-1ML) or ECL anti-rabbit IgG horseradish peroxidase (GENA934-1ML) from Sigma-Aldrich were used as secondary antibodies at 1:5000 dilution.

## Immunofluorescence staining

Cells were plated in 96-well plates (Perkin Elmer). For EdU pulse experiments in Fig. 1b or siRNA screen in Fig. 2a–c, cells were treated with 10 mM EdU (Jena Bioscience) for 30 min. Fixation was performed with 4% methanol-free PFA (Science Ser-vices) for 10 min at RT. Permeabilization was carried out with 0.5% w/v Triton X-100/PBS for 15 min at RT. Blocking was performed with 5% w/v BSA/PBS for 1 h at RT. For EdU pulse experiments, newly synthesized DNA was visualized by copper(I)-catalyzed azide-alkyne cycloaddition (100 mM Tris pH 8.5, 4 mM CuSO4, 10 mM AFDye 647 Azide (Jena Bioscience), 10 mM L-ascorbic acid). The cells were then incubated with the primary antibodies in blocking buffer overnight at 4 °C. After washing, cells were incubated with appropriate fluorophore-conjugated secondary antibody for 1 h at RT. Counterstaining for nuclei detection was performed by 2.5 mg/ml Hoechst 33342 (Sigma-Aldrich) for 10 min at RT. Images were acquired with an Operetta CLS High-Content Imaging System at 20×, 40× or 63× magnification using a water immersion objective. Images were processed and analyzed using Harmony High Content Imaging and Analysis Software (PerkinElmer) and R. The following primary antibodies were used: anti-KAP1 phospho-S824 (Abcam Cat# ab70369, RRID:AB_1209417, 1:600) and anti-Phospho-Histone H2A.X Ser139 (Cell Signaling Technology Cat# 2577, RRID:AB_2118010, 1:500). The following secondary antibodies were used: Alexa Fluor® 647 Goat anti-rabbit IgG (H + L) (Thermo Fisher Scientific Cat# A-21244, RRID:AB_2535812, 1:400) and Alexa Fluor® 488 Goat anti-rabbit IgG (H + L) (Thermo Fisher Scientific Cat# A-11008 (also A11008), RRID:AB_143165, 1:400).

## siRNA screen

RNAi cherry-pick library (SMARTpool, Horizon Discovery) was diluted according to the manufacturer's instructions. KPC cells were seeded and transfected at the same time in 96-well plates (PerkinElmer) using Lipofectamine RNAiMAX (Thermo Fisher Scientific) according to the manufacturer's instructions following reverse transfection method at a final concentration of 25 nM siRNA per well. Fixation was performed 48 h after transfection. EdU pulse and staining were performed as described in immunofluorescence staining. The percentage of positive cells was used to calculate the z-score, where $z = (x\text{-}\mu)/\sigma$ ($x$ is the percentage of positive cells in a certain well, $\mu$ is the percentage of positive cells in all wells and $\sigma$ is the standard deviation across all wells). Z-score was calculated to rank candidates according to their effect on increasing/decreasing the percentage of positive cells. $P$ values were calculated using unpaired $t$-test, two-sided for "EdU incorporation" and one-sided $t$-test for "-ATRi" and "+ATRi" readouts, with the likelihood of siRNA targeting a candidate to increase the percentage of pKAP1 positive cells than siNTC. $P < 0.05$ was used for considering a positive hit, except for "+ATRi" $P < 0.15$ was used. Raw data of the siRNA screen are presented in Supplementary Data 1.

## Proximity ligation assay

KPC cells expressing doxycycline-inducible shRNA targeting *MYC* were seeded in a 384-well format (PerkinElmer). Where indicated, cells were treated with doxycycline (1 mg/ml, 48 h) or equivalent amounts of ethanol. For the experiment in Fig. 1e, ice-cold methanol was used for fixation and permeabilization for 20 min at −20 °C. For the experiment in Fig. 3a, fixation and permeabilization was performed as mentioned in immunofluorescence staining. Cells were incubated with the indicated mouse and rabbit antibodies in blocking buffer overnight at 4 C. Cells were incubated with plus (Sigma-Aldrich, DUO92002) and minus (Sigma-Aldrich, DUO92004) probes directed against rabbit and mouse antibodies, respectively, for 1 h at 37 C. Washing with PLA Wash buffer A (Sigma-Aldrich) was followed by ligation for 30 min at 37 C. After washing with PLA Wash buffer A, in situ PCR amplification was then performed using Alexa 495 (Sigma-Aldrich, DUO92014) or 644 (Sigma-

Aldrich, DUO92013) conjugated oligonucleotides for 2 h at 37 C. Samples were counterstained with Hoechst 33342 (Thermo Fisher Scientific). Images were acquired using the Operetta CLS High-Content Analysis System at 40x magnification (PerkinElmer), processed and analyzed using Harmony High-Content Imaging and Analysis Software (PerkinElmer) and Prism v8.2.1. For Fig. 3a, background spots with low intensity were discarded from the analysis. The following primary antibodies were used: anti-total RNAPII (Santa Cruz Biotechnology Cat# sc-55492, RRID:AB_630203, 1:1500), anti-PCNA (Abcam Cat# ab92552, RRID:AB_10561973, 1:1500), anti-RAD9 (Thermo Fisher Scientific Cat# PA5-21275, RRID:AB_11152972, 1:500), anti-RNAPII phospho Ser5 (BioLegend Cat# 904001, RRID:AB_2565036, 1:500), anti-CTR9 (Novus Cat# NB100-68205, RRID:AB_1108024, 1:500).

### RNA extraction and real-time quantitative PCR (RT−qPCR)

Total RNA was isolated using peqGOLD TriFast (PeqLab). Reverse transcription was performed using 1−2 μg RNA template, random primers, dNTPs, MLV buffer and MLV reverse transcriptase. cDNA was then used for quantitative PCR (qPCR) using the appropriate primers and PowerUP SYBR Green Master Mix (Thermo Fisher Scientific). For mRNA qPCRs, all conditions were normalized to the corresponding mRNA of the gene *Vcl*. All primer sequences are shown in Supplementary Data 3.

### RNA sequencing

KPC cells expressing doxycycline-inducible shRNA targeting CTR9, CDC73, MYC or luciferase as a control were incubated with doxycycline (1 mg/ml) for 48 h. RNA was isolated using peqGOLD TriFast (PeqLab). Total RNA was extracted using the miRNeasy kit (QIAGEN) including on-column DNase I digestion. mRNA was isolated using the NEBNext Poly(A) mRNA Magnetic Isolation Module (NEB) and library preparation was performed using the NEBNext Ultra II Directional RNA Library Prep Kit for Illumina according to the instructions. Libraries were size selected using Agentcourt AMPure XP Beads (Beckman Coulter), followed by amplification with 11 cycles of PCR.

### Chromatin-based high-throughput sequencing

ChIP and ChIP-Rx (ChIP-sequencing) with spike-in normalization were performed as previously described[22], with the slight modification of using sonication (20 min, 25% amplitude, 10 s on, 30 s off) to fragment the DNA prior to immunoprecipitation. CUT&RUN followed by sequencing was based on ref. 70 When specified, 10% U2OS cells were added as spike-in. The following primary antibodies were used for ChIP (3 μg) or ChIP-sequencing (15 μg): anti-RNAPII (Santa Cruz Biotechnology Cat# sc-17798, RRID:AB_677355), anti-SPT6 (Novus Cat# NB100-2582, RRID:AB_2196402, 1.5 μg for ChIP and 10 μg for ChIP-sequencing), anti-SPT5 (Santa Cruz Biotechnology Cat# sc-133217, RRID:AB_2196394), anti-RTF1 (Bethyl Cat# A300-179A, RRID:AB_2185963), anti-RNAPII phosphor-Ser2 (Abcam Cat# ab5095, RRID:AB_304749), anti- RNAPII phosphor-Ser5 (BioLegend Cat# 904001, RRID:AB_2565036), anti-SPT4 (Cell Signaling Technology Cat# 64828, RRID:AB_2756442, 1.5 μg for ChIP), anti-CTR9 (Bethyl Cat# A301-395A, RRID:AB_960973). All libraries were sequenced on the NextSeq2000 Illumina platform, paired-end sequencing with 60 cycles for read1 and 60 cycles for read2, except for BLISS libraries that were sequenced on NextSeq2000, single-read sequencing with 101 cycles for read1.

### BLISS

BLISS was performed as described in ref. 22. Cells were fixed with 3.7% v/v paraformaldehyde and washed with PBS. Lysis was performed by incubation in lysis buffer 1 (10 mM Tris-HCl, 10 mM NaCl, 1 mM EDTA, 0.2% Triton X-100, pH 8) for 1 h at 4 C, followed by a brief rinse in PBS and incubation in lysis buffer 2 (10 mM Tris-HCl, 150 mM NaCl, 1 mM EDTA, 0.3% SDS, pH 8) for 1 h at 37 C. After rinsing in PBS, cells were equilibrated in CutSmart buffer (New England Biolabs) before double-

strand breaks were blunted using the Quick Blunting Kit (New England Biolabs) according to the manufacturer's protocol. After PBS rinse and before blunting, AsiSi (New England Biolabs) digestion was performed according to the manufacturer's instructions. Sense and antisense adaptor oligos were annealed by heating at 95 °C for 5 minutes, followed by a gradual cooling to 25 °C over a period of 45 min. Following equilibration in CutSmart buffer (New England Biolabs) and T4 Ligase buffer (New England Biolabs), annealed adapters were applied to samples and ligated with T4 DNA Ligase (New England Biolabs) for 16 h at 16 °C according to the manufacturer's recommendations. Excess adapters were removed by repeated rinses in high salt wash buffer (10 mM Tris-HCl, 2 M NaCl, 2 mM EDTA, 0.5% Triton X-100, pH 8). Genomic DNA was extracted in DNA extraction buffer (1% SDS, 100 mM NaCl, 50 mM EDTA, 10 mM Tris-HCl, pH 8) supplemented with Proteinase K (1 mg/ml, Roth) for 16 h in a thermoshaker at 55 °C. DNA was isolated by phenol-chloroform extraction and isopropanol precipitation, resuspended in TE buffer and sonicated with a Covaris Focused Ultrasonicator M220 for 1 to 2 min to obtain a fragment size of 300−500 bp. Fragment size was determined on the Fragment Analyzer (Agilent) using the NGS Fragment High Sensitivity Analysis Kit (1−6000 bp; Agilent). DNA was concentrated using Agencourt AMPure XP Beads (Beckman Coulter), transcribed into RNA, and DNA digested using the MEGAscript T7 Transcription Kit (Thermo Fischer Scientific) according to the manufacturer's recommendations. RNA purification was performed using Agencourt RNAClean XP Beads (Beckman Coulter). RNA concentration was determined on the Fragment Analyzer (Agilent) using the Standard Sensitivity RNA Analysis Kit (Agilent). Library preparation was performed by ligation of the RA3 adaptor to samples using T4 RNA Ligase 2 (New England Biolabs) supplemented with recombinant ribonuclease inhibitor (Thermo Fischer Scientific). Samples were reverse transcribed using SuperScript III Reverse Transcriptase Kit (Thermo Fisher Scientific), and library indexing and amplification were performed using NEBNext High-Fidelity 2X PCR Master Mix (New England Biolabs) with RP1- and desired RPI-primer. Libraries were purified using Agentcourt AMPure XP Beads (Beckman Coulter), quality, quantity, and fragment size assessed on the Fragment Analyzer (Agilent) using the NGS Fragment High Sensitivity Analysis Kit (1−6000 bp; Agilent), and then subjected to Illumina sequencing according to the manufacturer's instructions. Adapters and oligos were custom synthesized, and unique molecular identifiers (UMIs) were generated by random incorporation of the four standard dNTPs using the 'Machine mixing' option.

### 4sU sequencing

4sU sequencing were performed as previously described[21]. Subconfluent cells were treated as described. During the last 15 min of treatment, cells were incubated with 2 mM 4-thiouridine (4sU; Sigma-Aldrich) before lysis with QIAzol reagent (QIAGEN). Total RNA extraction was performed using the miRNeasy kit (QIAGEN) with on-column DNase digestion, and RNA quantity and quality were assessed using a Nanodrop spectrophotometer and a Fragment Analyzer (Thermo Fisher Scientific), respectively. 20−40 μg of RNA with RQN values greater than 9 was then biotinylated by adding EZ-Link Biotin-HPDP (Pierce) in 0.2 mg/ml dimethylformamide and biotin labeling buffer (10 mM Tris pH 7.4, 1 mM EDTA) and incubating for 2 h at RT with rotation. Biotinylated RNA was then purified by chloroform isoamyl alcohol extraction in MaXtract high-density tubes (QIAGEN). The upper aqueous phase was collected and 1/10 volume of 5 M NaCl and an equal volume of isopropanol were added. All samples were centrifuged at 20,000 g for 20 min at 4 °C, the RNA pellets were washed with 75% ethanol and centrifuged at 20,000 g for 10 min at 4 °C. After drying, the pellets were resuspended in nuclease-free water. Biotinylated RNA samples were then eluted using Dynabeads MyOne Streptavidin T1 beads (Invitrogen) in Dynabeads Wash Buffer (2 M NaCl, 10 mM Tris pH 7.5, 1 mM EDTA, 0.1% v/v Tween 20) for 15 min at RT

with rotation. After stringent washing, 4sU-labeled RNA was eluted from the beads with 100 ml of freshly prepared 100 mM DTT in nuclease-free water and purified using the RNeasy MinElute Kit (QIAGEN). Samples were quantified using the RiboGreen Assay (Invitrogen), and 5–20 ng RNA was used for cDNA library preparation by first depleting rRNA using the NEBNext rRNA Depletion Kit (Human/Mouse/Rat) (New England Biolabs) and then the NEBNext Ultra II Directional RNA Library Prep Kit (New England Biolabs). cDNA libraries were amplified using appropriate multiplexed index primers in 11–14 cycle PCR, depending on the amount of RNA input.

## Bioinformatical analysis and statistics

ChIP-sequenced libraries were mapped separately to mouse mm10 genome and human hg19 (spike-in) using Bowtie 2 v2.3.4.1[71] with default parameters. ChIP-Rx spike-in normalization was performed by calculating spike-in normalization factor, by dividing the number of mapped reads from the spike-in of the smallest sample by the number of mapped reads from the spike-in of each sample. For each sample, this factor was multiplied by the number of reads that mapped to the mouse genome. Samtools[72] was used to manipulate the bam files (indexing and subsampling). To visualize read alignments on the Integrated Genome Browser v9.1.10, BAM files were converted to bigwig files using deepTools v3.5.1 with no further normalization and a bin size of 10 bases. Heat map (Supplementary Fig. 4c) was generated by applying the computeMatrix operation to all expressed genes (at least 10 reads mapped per gene based on mRNA-sequencing) followed by plotHeatMap function in deepTools. Heat maps (Fig. 5e, Supplementary Fig. 7a) were generated using bamCompare function first followed by plotHeatMap in Deptools.

CUT&RUN libraries were mapped to the mouse mm10 genome using default parameters using Bowtie 2 v2.3.4.1. Samples were read-normalized with the number of reads in the smallest sample. BAM files were converted to bigwig files using deepTools with no further normalization and a bin size of 10 bases. Metagene plot (Fig. 3c) was generated by applying the computeMatrix operation to all expressed genes (Centered around TSS) followed by plotProfile function in deepTools.

For mRNA-seq, reads were mapped to mm10 using Bowtie 2 v2.3.4.1 then samples were normalized to the number of mapped reads in the smallest sample. Reads per gene were counted using the featureCounts function from the R package Rsubread. Non- and low-expressed genes were removed with less than 5 reads per gene. Differentially expressed genes relative to control cells expressing shRNA targeting luciferase, were retrieved using edgeR and $P$ values were adjusted for multiple testing using the Benjamini–Höchberg procedure. Gene set enrichment analysis[73] was done with the Hallmark or the C5 (ontology gene sets) databases from MSigDB[74]. GO terms were identified using GOstats and enrichR packages in R[75]. For the table in Fig. 3e, threshold of log2FC of <−1 was set for the GO term search. In Supplementary Fig. 5b, differential gene expression was calculated based on reads mapping only to the second half of the gene. In Supplementary Fig. 5d, down-regulated genes in the DNA repair GO term (GO-0006281) were selected based on a $\log_2 FC < 0$ and FDR < 0.05.

For BLISS, samples were demultiplexed based on their condition-specific barcodes using UMI-tools v1.0.1[76], allowing for one mismatch in the barcode, and mapped to the mouse reference genome (mm10) using Bowtie2 v2.3.4.1. For normalization, AsiSI-specific restriction sites were generated by in silico digestion of the mm10 genome, followed by counting deduplicated reads in AsiSI-specific restriction sites using countBamInGRanges from the R package exomeCopy. The sample with the lowest number of AsiSI-specific reads was divided by the number of corresponding reads from each sample. The resulting ratio was multiplied by the total number of deduplicated reads, and samples were then randomly subsampled to the calculated number of

reads. For Supplementary Fig. 1b, AsiSI-normalized mapped reads were annotated to the respective described genomic region using ChIP-seeker v1.34.1 in R v4.2.2.

For the analysis of the 4sU sequencing experiment, reads were mapped to the mouse reference genome (mm10) with Bowtie 2 v2.3.4.1. Reads mapping to exons and regions in the ENCODE Blacklist[77] were removed from BAM files using bedtools v2.26.0. Samples were then normalized by the bamCoverage function in deepTools using counts per million mapped reads, and then bigwig files were generated. For the metagene quartile plots in Fig. 3g and Supplementary Fig. 7b, expressed genes were stratified based on their length from quartile 1 (shortest) to quartile 4 (longest), where each quartile consists of 2091 genes, except quartile 4 with 2092 genes. Since only intronic 4sU reads were considered; thus, genes shorter than 10 kb were excluded from the analysis, as very short genes have a very low number of reads after filtering for exons.

The data shown in Supplementary Fig. 3f, g were generated using R and data were obtained from two databases: CRISPR (DepMap Public 23Q2 + Score, Chronos) and OmicsExpressionProteinCodingGenesTPMLogp1. The CRISPR database contains gene dependency values for each cell line, while the OmicsExpressionProteinCodingGenesTPMLogp1 database stores gene expression profiles for each cell line. Both datasets were obtained from DepMap[32]. The axes labels are defined by DepMap[32]. Data filtering was performed exclusively for pancreatic carcinoma cell lines.

For statistical analysis, the statistical tests for the results shown in each panel are described in the figure legends. All statistical analyses were performed using either Prism 9.5.1 or R v4.2.2.

## Reporting summary

Further information on research design is available in the Nature Portfolio Reporting Summary linked to this article.

## Data availability

ChIP-Rx, BLISS and RNA sequencing data are available from the Gene Expression Omnibus (https://www.ncbi.nlm.nih.gov/geo/) under accession number GSE228800, including GSE228795 and GSE243703 (ChIP-seq/CUT&RUN), GSE228797 (RNA-seq), GSE228799 (BLISS) and GSE243701 (4sU-seq). The mm10 (GRCm38) and hg19 (GRCh37.p13) reference genomes are available at https://www.ncbi.nlm.nih.gov/datasets/genome/GCF_000001635.20/ and https://www.ncbi.nlm.nih.gov/datasets/genome/GCF_000001405.25/, respectively. The mm10 blacklisted regions are available at https://www.encodeproject.org/files/ENCFF543DDX/. Source data are provided with this paper. The remaining data are available within the Article, Supplementary Information or Source Data file. Source data are provided with this paper.

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

## Acknowledgements
This work was supported by grants from the European Research Council (SENATR 101096948), the Excellence Program of the German Cancer Aid (#70114538) and the German Research Foundation (EI 222/21-1 and INST 93/1023-1-FUGG). The authors thank Tobias Roth, Barbara Bauer, Ulrike Samfaß, Ryan Ramjan, and Sarah Hess for technical support.

## Author contributions
A.G. performed most tissue culture experiments and A.G.-W performed all in vivo experiments. B.K., G.M., M.J. and A.R. performed and analyzed flow cytometry experiments. M.R., A.G., A.G.-W analyzed IHC. C.S.-V measured high-content immunofluorescence experiments. C.P.A. measured all sequencing experiments and A.G., P.G., R.S.V. and C.P.A. analyzed all sequencing data. F.M. performed the dependency analysis in Supplementary Fig. 3f,g. B.Kn. provided help with the in vivo experiments. B.K., G.C., S.A, C.K., performed additional experiments and provided scientific input. G.G. helped to interpret the flow cytometry data. M.E. and A.G. conceptualized the study. M.E. devised and supervised experiments and wrote the paper with input from all authors.

## Funding

## Competing interests
M.E. is a founder and shareholder of Tucana Biosciences. The remaining authors declare no competing interests.
