## [Peer Review File · Nature Communications]

REVIEWER COMMENTS

Reviewer #1 (Remarks to the Author):

The authors do an adequate job responding to the reviewers' comments, still for this reviewer there are a number of comments that were not addressed or have come to light based on the newer, better version of this manuscript, particularly around the studies of DNA repair and immunology is concerned.

The DNA damage assays performed are still rather superficial in nature and without overt damage being induced, so the changes in for instance H2Ax may be due to background damage in culture. Additionally, in the drug profiling work, no relevant DNA damage drugs are being used (albeit PARP inhibitor-which is a sensitizer). E.g., the investigators should think about FOLFIRINOX and GEM the standard of care regimens for PDAC.

Additionally, the conclusions and schematic made about the tumor immunology new data is intriguing and promising, but lacks appropriate modeling. Can they perform these studies in a matched background, with an endogenous antigen (OVA), with depletion studies?

Finally, the addition of one human PDAC line for one of these studies does not provide overall rigor for the work concluded on. Again, I would like to support this work and applaud the investigators for their additional work and work on the manuscript, however the conclusions and interpretations just might be slightly premature for this journal without additional data supplied.

Reviewer #2 (Remarks to the Author):

In this revised manuscript, Gaballa et al. provide additional support linking MYC and PAFc to S-phase progression, DNA damage control, and immune surveillance in pancreatic ductal adenocarcinoma models. The paper makes several important contributions, including demonstration that depletion of PAFc subunits (1) decreases growth of PDAC cells in culture and tumors in a mouse model, (2) increases DNA damage as indicated by elevated levels of pKAP1- and gamma-H2AX-positive cells, and (3) increases expression of genes involved in immunosuppression of tumor cells (MHC genes) while decreasing expression of genes involved in DNA damage repair. The authors connect these biological effects to PAFc-dependent effects on transcription elongation: long genes, including those that encode DNA repair proteins, are down-regulated upon CTR9 depletion and short genes, such as MHC genes, are up-regulated. This is an interesting result that relates the functions of a global positive elongation factor to a biological outcome. The authors also provide strong support for their hypothesis that CTR9 is required for PDAC tumor growth by restricting immune cell infiltration.

The authors have done a nice job revising the manuscript to address previous reviewer comments. Modifications include the presentation of new experiments (e.g. the use of additional PDAC cell lines, the testing of CTR9 depletion effects in an inducible MYC-ER cell line, and the inclusion of additional data replicates with statistical analysis), more informative analyses of genomic data, and revisions to the text that appropriately soften some of the conclusions. Overall, this is a high quality, interesting study that links the positive elongation functions of PAFc to gene expression profiles that have consequences for DNA damage control and immune response in a cancer model. A few issues remain.

1. Previous reviewers noted that the connections between MYC and PAFc are not completely clear in the manuscript, and this is still the case. Although the authors arrived at PAFc by screening MYC-related regulatory proteins for phenotypes similar to those caused by MYC depletion, a path from MYC to its cellular outcomes need not be entirely through PAFc. I feel the authors can be more careful in their writing. For example, in Figure S3B, the authors show that either activation of MYC-ER or knockdown of CTR9 increases DNA damage. While they state in line 163 that MYC-ER “exacerbated DNA damage responses to depletion of CTR9”, this effect is not evident for pKAP1, which is routinely used as a metric of DNA damage in the paper. Moreover, if signaling from MYC to DNA damage response goes through PAFc, then why would additive effects be expected? In addition, the reader would benefit from an explanation of why both MYC activation (Figure S3B), which would be expected to increase PAFc recruitment to Pol II and expression of DNA repair genes, and MYC depletion (Figure 1), which would be expected to decrease PAFc recruitment and expression of DNA repair genes, increase DNA damage (as measured by pKAP1).
2. Figure S1A, left. Are all the cells labeled in red because they are gamma-H2AX positive?
3. Figure S3B. Most of the proteins shown in the western are not discussed in the paper or in the legend. An explanation of their importance to the experiment would be helpful.
4. Figure 3B. Only a subset of the genes being tested are mentioned in the paper. Some information on why TCEB3 and other genes are included would be helpful.
5. Line 161. The wording needs to be corrected.
6. Lines 334-335. The wording needs to be corrected.

Reviewer #4 (replacing Reviewer #3, Remarks to the Author):

I have extensively gone through the raised concerns and critiques by Reviewer #3 and checked whether the authors have adequately addressed them.

Based on this, my assessment is that the authors have added more experimental evidence and clarifications in the revised manuscript and therefore adequately addressed the concerns from all 3 reviewers in an appropriate manner. The results as presented now are convincing and support the major conclusions and therefore I am supportive of publication of the manuscript in Nature Communications.

Reviewer #5 (Remarks to the Author):

In this study, Gaballa and colleagues analyzed the mechanisms that link MYC-induced DNA damage during S-phase progression to anti-tumor immune evasion in pancreatic ductal adenocarcinoma (PDAC). They show that the PAF1c transcription elongation complex is critical for the prevention of DNA damage during S-Phase progression – particularly in cells with high MYC expression – by promoting transcription elongation of several long genes that are involved in DNA replication and repair. Intriguingly, these DNA replication and repair genes compete with MHC class I genes for PAF1c, thereby impacting on the anti PDAC immune response. Accordingly, the authors demonstrate that PAF1c is required for PDAC maintenance in vivo. This is not due to the role of PAF1c in limiting DNA damage, but MHC-I dependent activation of T-cells. Depletion of the PAF1c subunit CTR9 releases RNA polymerase and elongation factors from long genes, such as the ones required for DNA replication and repair and promotes transcription of short genes including MHC class I. Thus, PAF1c is a critical rheostat in PDAC, with is required for the prevention of DNA damage and MYC-driven immune evasion.

Overall, this is a very comprehensive and detailed study with high clinical relevance elucidating a critical role of PAF1c in PDAC and its link to MYC, S-phase progression and immune evasion. The data as presented are sound, robust and of high significance for the field. Particularly, the demonstration of how CTR9 depletion causes upregulation of MHC class I gene expression and activation of T-cells is very intriguing and of translational relevance considering that PDAC is an “immunologically cold” tumor largely resistant to immunotherapies.

Specific comments:

- 1) MYC has been shown to mediate immune evasion by different mechanisms and the authors demonstrate that TBK1 is activated upon depletion of MYC but not CTR9. This suggests that MYC and CTR9 might suppress the immune system via distinct mechanisms? Thus, it would be of great interest to compare the consequences of MYC and CTR9 depletion on the expression immunomodulatory factors (cytokines, interferons, immune checkpoints) and the composition of the immune landscape of PDAC. Thereby, it is possible to better understand the specific effects of MYC and CTR9 depletion on PDAC immunity.
- 2) The authors state that „PAF1c is required to limit DNA damage in cells expressing high MYC levels” (page 8, line 170), which they demonstrate using the MYC-ER model. Do PDAC cells with high MYC expression also show higher expression of PAF1c and are they more susceptible to CTR9/CDC73 depletion (for example in CRISPR Screens)?
- 3) In Figure 4f, mice with shCTR9 display a markedly longer survival than shMYC mice. How do the author interpret this finding?
- 4) In Figure 5a and Supplementary Figure 6b, the authors show that depletion of CTR9 and CDC73 caused an increase in expression of MHC class I genes and they conclude: “This caught our attention, since suppression of MHC class I gene expression is a major mechanism of MYC-mediated immune evasion^{40, 41} and the observation suggested that depletion of CTR9, like that of MYC, might render tumors visible for the host immune system.” And later in the discussion (page17, line 430/431), the authors write that “Our data suggest that MYC-dependent immune evasion is a direct consequence of its effects on transcriptional elongation.”

It would be important to confirm this statements by additional experiments. For example, by showing that suppression of MHC class I expression upon MYC overexpression (by MYC-ER) can be overcome by depletion of CTR9.

5) In the discussion, the authors propose “that long and short genes compete for components of the basal transcriptional machinery and that this competition is a central mechanism of MYC-dependent immune evasion”.

Could the authors show whether this shift from transcription of long to short genes upon depletion of CTR9 occurs also upon depletion of MYC?

Reply to Referees

We would like to thank all reviewers for their efforts and for their thoughtful comments. Please find below our replies

Reviewer #1 (Remarks to the Author)

„The DNA damage assays performed are still rather superficial in nature and without overt damage being induced, so the changes in for instance H2Ax may be due to background damage in culture.“

There can be no doubt from thousands of papers that there are multiple sources of endogenous DNA damage, and that all organisms have developed elaborate DNA repair and checkpoint mechanisms to cope with them. All these responses are studied mechanistically in cultured cells, so it is uncertain what the term “background damage in culture” refers to. Here we study one of the myriad mechanisms that allow cells to cope with one potential source of DNA damage – that associated with DNA replication. We show that interference with this mechanism induces DNA damage in culture and *in vivo*, hence it is not an artefact.

“Additionally, in the drug profiling work, no relevant DNA damage drugs are being used (albeit PARP inhibitor-which is a sensitizer). E.g., the investigators should think about FOLFIRINOX and GEM the standard of care regimens for PDAC.“

All the drugs that we use have a defined mechanism of action and hence allow clear conclusions about the DNA damage response pathways in which the pathway we describe is relevant. The aim of the study was not to mimic the response of cells to clinically used chemotherapeutics with their complex modes of action.

“Additionally, the conclusions and schematic made about the tumor immunology new data is intriguing and promising but lacks appropriate modeling. Can they perform these studies in a matched background, with an endogenous antigen (OVA), with depletion studies?“

We would argue that the modeling of the immune responses with an OVA system is interesting, but I think the reviewer will agree that it also constitutes an entire study in itself.

“Finally, the addition of one human PDAC line for one of these studies does not provide overall rigor for the work concluded on.“

We document the PAF1-dependent mechanisms and their therapeutic value in both murine and human cell lines and *in vivo* hence we would argue that the mechanism and its relevance are rigorously documented. We have added RQ-PCR assays showing the differential effect of CTR9 depletion on replication genes and MHC class I expression in another human cell line, PANC1 (Supplementary Figure 6h). In reply to this and to comment #2 of reviewer 5, we have also included a bioinformatic analysis showing that the dependence of multiple human PDAC cell lines on multiple PAF1c complex subunits is highly correlated with levels of MYC and MYC target gene expression (Supplementary Figure 3f) and that this is highly specific for subunits of PAF1c (Supplementary Figure 3g)(described in more detail in the reply to reviewer 5).

Reviewer #2

1. „Previous reviewers noted that the connections between MYC and PAFc are not completely clear in the manuscript, and this is still the case. Although the authors arrived at PAFc by screening MYC-related regulatory proteins for phenotypes similar to those caused by MYC depletion, a path from MYC to its cellular outcomes need not be entirely through PAFc. I feel the authors can be more careful in their writing.

We have now explained the underlying model more carefully in the discussion (line 400-413). It is very clear from this and many other studies that MYC has multiple functions in controlling RNAPII function that many are independent of PAF1: for example, MYC activates genes that promote cell growth (e.g. Figure 3d), and it promotes loading of RNAPII to its target genes. This study reveals that PAF1 protects from the DNA damage and from the immune recognition associated with MYC-driven proliferation. Our view is – and that is shown by the data – that depletion of PAF1 selectively disables a genome-protective function of MYC and hence opens a wide therapeutic window, as witnessed by the survival data. A detailed discussion of the model is given in (Papadopoulos *et al*, 2023), which is quoted in the text.

For example, in Figure S3B, the authors show that either activation of MYC-ER or knockdown of CTR9 increases DNA damage. While they state in line 163 that MYC-ER “exacerbated DNA damage responses to depletion of CTR9”, this effect is not evident for pKAP1, which is routinely used as a metric of DNA damage in the paper.”

To address this comment, we have replaced the pKAP1 blot with one that more clearly documents the effect (Supplementary Figure S3b). To obtain quantitative data, we have added an evaluation of pKAP1 and gH2Ax staining by immunofluorescence as a new Supplementary Figure S3c. Both results document that activation of MYCER exacerbates the DNA damage in response to depletion of CTR9.

“Moreover, if signaling from MYC to DNA damage response goes through PAFc, then why would additive effects be expected? In addition, the reader would benefit from an explanation of why both MYC activation (Figure S3B), which would be expected to increase PAFc recruitment to Pol II and expression of DNA repair genes, and MYC depletion (Figure 1), which would be expected to decrease PAFc recruitment and expression of DNA repair genes, increase DNA damage (as measured by pKAP1).”

The reply is very similar to the comments above. MYC promotes cell proliferation and – since proliferating cells have more damage than resting cells – activation of MYC increases DNA damage. PAF1 is required to protect cells from excessive DNA damage and hence has a specific protective role in cells that express high levels of MYC and are highly proliferative.

“2. Figure S1A, left. Are all the cells labeled in red because they are gamma-H2AX positive?”

We apologize for using the same color used for yH2AX and have now used blue color to mark S-phase cells.

“3. Figure S3B. Most of the proteins shown in the western are not discussed in the paper or in the legend. An explanation of their importance to the experiment would be helpful.”

In reply to the comment, we have removed non-essential blots of markers of DNA damage (RPAS33 and RPAS4/S8) and have added an explanation in the text (line 166/167) for the genes that we describe in Supplementary Figure S3B and S3C.

“4. Figure 3B. Only a subset of the genes being tested are mentioned in the paper. Some information on why TCEB3 and other genes are included would be helpful.”

We have added an explanation for all genes that we tested (line 202ff).

“5. Line 161. The wording needs to be corrected.”

“6. Lines 334-335. The wording needs to be corrected.”

We have corrected the wordings.

Reviewer #4

No comments that need to be addressed.

Reviewer #5

“1) MYC has been shown to mediate immune evasion by different mechanisms and the authors demonstrate that TBK1 is activated upon depletion of MYC but not CTR9. This suggests that MYC and CTR9 might suppress the immune system via distinct mechanisms? Thus, it would be of great interest to compare the consequences of MYC and CTR9 depletion on the expression immunomodulatory factors (cytokines, interferons, immune checkpoints) and the composition of the immune landscape of PDAC. Thereby, it is possible to better understand the specific effects of MYC and CTR9 depletion on PDAC immunity.”

We have now added a panel that directly compares the changes in gene expression upon depletion of MYC and of CTR9/CDC73 and include this as a new Supplementary Figure 6f. This shows that the effects of all three on DNA replication genes are very similar, but the effects on immunomodulatory factors are very different. Specifically, multiple gene sets of interferon (IRF-dependent) and cytokine (NF- κ B-dependent) genes are upregulated upon depletion of MYC, but not upon depletion of CTR9 or CDC73, consistent with the fact that IRFs and NF- κ B are downstream targets that are activated by TBK1.

„2) The authors state that „PAF1c is required to limit DNA damage in cells expressing high MYC levels” (page 8, line 170), which they demonstrate using the MYC-ER model. Do PDAC cells with high MYC expression also show higher expression of PAF1c and are they more susceptible to CTR9/CDC73 depletion (for example in CRISPR Screens)?“

We have now included the requested analysis. The results show unequivocally that the dependence of multiple human PDAC lines on the CTR9, CDC73 and LEO1 subunits of PAF1c is highly correlated both with high levels of MYC and with high MYC target gene expression (Supplementary Figure 3f). This is highly specific since the dependence of PDAC cells on other genes that were identified in the screen is not correlated with MYC levels (Supplementary Figure 3g).

„3) In Figure 4f, mice with shCTR9 display a markedly longer survival than shMYC mice. How do the author interpret this finding?“

We thank the reviewer for pointing this out. When we depleted MYC, we noted that all recurring tumors restore the expression of MYC target genes. Some of them simply silenced the shRNA; but another group of tumors responded by adapting the network of proteins that interact with the MYC partner protein MAX. This has two arms: MYC/MAX complexes activate gene expression, whereas MXD/MAX complexes repress transcription, many of the recurring tumors upon MYC depletion simply silence MDX genes and hence restore activation of target genes. Hence the dynamic nature of this network makes it much easier to restore MYC function (see line 426 ff).

„4) In Figure 5a and Supplementary Figure 6b, the authors show that depletion of CTR9 and CDC73 caused an increase in expression of MHC class I genes and they conclude: “This caught our attention, since suppression of MHC class I gene expression is a major mechanism of MYC-mediated immune evasion 40, 41 and the observation suggested that depletion of CTR9, like that of MYC, might render tumors visible for the host immune system.” And later in the discussion (page17, line 430/431), the authors write that “Our data suggest that MYC-dependent immune evasion is a direct consequence of its effects on transcriptional elongation.”It would be important to confirm this statement by additional experiments. For example, by showing that suppression of MHC class I expression upon MYC overexpression (by MYC-ER) can be overcome by depletion of CTR9.“

We have added a panel as Supplementary Figure 6d documenting the expression of two major MHC class I genes, H2-D1 and H2-K1. This shows a side-by-side comparison documenting that depletion of

CTR9 increases the expression of both genes more strongly than depletion of MYC, despite the very effective depletion of MYC (Figure 1a). This result corroborates the RNAseq data shown in Figure 5a. Importantly, depletion of CTR9 increased the expression of H2-D1 and H2-K1 in KPC-MYCER cells both in the presence of endogenous MYC levels and when MYCER was activated by OHT (Supplementary Figure 6d). The data show that the high endogenous MYC levels in KPC cells suppress MHC class I expression and that depletion of CTR9 overcomes this suppression even when MYC is further activated. We did not observe that activation of MYCER on top of the high endogenous MYC levels in KPC cells further reduces MHC class I expression.

„5) In the discussion, the authors propose “that long and short genes compete for components of the basal transcriptional machinery and that this competition is a central mechanism of MYC-dependent immune evasion”. Could the authors show whether this shift from transcription of long to short genes upon depletion of CTR9 occurs also upon depletion of MYC?”

As we and others have shown in many papers and systems, MYC has many effects on RNAPII function and dynamics, and both effects on elongation and RNAPII loading have been reported, with the detailed of both effects varying between systems and experimental details. In the KPC cells used here, the dominant effect of MYC depletion is a strong global decrease in RNAPII chromatin association. This makes it much harder to detect a specific shift in elongation. A detailed analysis shows that RNAPII association with DNA repair genes decreases strongly and that MHC class I genes are protected from this consistent with the effects on gene expression. We are happy to include these data, but we feel that the data that we describe above in response to the first comment of this reviewer and that we include as Supplementary Figure 6f highlight the relevant differences between MYC and CTR9/CDC73 depletion in a much more impactful and biologically relevant manner.

Reference

Papadopoulos D, Uhl L, Ha SA, Eilers M (2023) Beyond gene expression: how MYC relieves transcription stress. *Trends Cancer* 9: 805-816

REVIEWERS' COMMENTS

Reviewer #1 (Remarks to the Author):

The authors do an adequate job of addressing the reviewers' comments and I appreciate the authors response. Still, some comments are not directly addressed (both experimentally with rigor), but as the authors do point out many of these points could be beyond the scope of this current work and they do provide a lot of interpretable data.

Reviewer #2 (Remarks to the Author):

The authors have satisfactorily addressed my previous concerns. I have only a few minor suggestions.

1. The new Supplemental Figures 3f and 3g are nice additions to the paper and make the case that Myc expression levels correlate with dependency of pancreatic cancer cells on PAFc. However, the current axes labels are vague and the units are undefined, making the data difficult for the reader to analyze.
2. Line 186: "break" instead of "beak"
3. Line 312: remove the word "data" after CHIP-sequencing
4. Line 452: replace "deletion" with "depletion"

Reviewer #5 (Remarks to the Author):

I appreciate the effort the authors have spent on addressing the comments of all reviewers. The revised manuscript has substantially been improved and my concerns have been fully addressed.

I would like to congratulate the authors for this highly relevant and elegant study that provides novel and translational relevant insights into pancreatic cancer biology.

Response to the reviewers' comments

We thank all three reviewers for their helpful comments.

Reviewer #1

No further changes are necessary.

Reviewer #2

„The new Supplemental Figures 3f and 3g are nice additions to the paper and make the case that Myc expression levels correlate with dependency of pancreatic cancer cells on PAFc. However, the current axes labels are vague and the units are undefined, making the data difficult for the reader to analyze.“

Regarding the axis labels in Supplementary Figures 3f and 3g, the labels that we used are defined by DepMap and we have added a statement and the relevant reference to the Methods section (line 834).

We have also corrected the spelling mistakes.

Reviewer #5

No further changes are necessary.